# Comparative Genomics and Transcriptomics Analyses Reveal a Unique Environmental Adaptability of *Vibrio fujianensis*

**DOI:** 10.3390/microorganisms8040555

**Published:** 2020-04-13

**Authors:** Zhenzhou Huang, Keyi Yu, Yujie Fang, Hang Dai, Hongyan Cai, Zhenpeng Li, Biao Kan, Qiang Wei, Duochun Wang

**Affiliations:** 1National Institute for Communicable Disease Control and Prevention, Chinese Center for Disease Control and Prevention (China CDC), State Key Laboratory of Infectious Disease Prevention and Control, Beijing 102206, China; ronnie_0414@163.com (Z.H.); yky0414@163.com (K.Y.); daihang@icdc.cn (H.D.); caihongyan@icdc.cn (H.C.); lizhenpeng@icdc.cn (Z.L.); kanbiao@icdc.cn (B.K.); 2Center for Human Pathogenic Culture Collection, China CDC, Beijing 102206, China; 3Center for Infectious Disease Research, School of Medicine, Tsinghua University, Beijing 100084, China; fangyujie2015@163.com; 4Office of Laboratory Management, China CDC, Beijing 102206, China

**Keywords:** *Vibrio fujianensis*, comparative genomics, transcriptomics, environmental adaptability, cross-agglutination reaction, salt tolerance

## Abstract

The genus *Vibrio* is ubiquitous in marine environments and uses numerous evolutionary characteristics and survival strategies in order to occupy its niche. Here, a newly identified species, *Vibrio fujianensis*, was deeply explored to reveal a unique environmental adaptability. *V. fujianensis* type strain FJ201301^T^ shared 817 core genes with the *Vibrio* species in the population genomic analysis, but possessed unique genes of its own. In addition, *V. fujianensis* FJ201301^T^ was predicated to carry 106 virulence-related factors, several of which were mostly found in other pathogenic *Vibrio* species. Moreover, a comparative transcriptome analysis between the low-salt (1% NaCl) and high-salt (8% NaCl) condition was conducted to identify the genes involved in salt tolerance. A total of 913 unigenes were found to be differentially expressed. In a high-salt condition, 577 genes were significantly upregulated, whereas 336 unigenes were significantly downregulated. Notably, differentially expressed genes have a significant association with ribosome structural component and ribosome metabolism, which may play a role in salt tolerance. Transcriptional changes in ribosome genes indicate that *V. fujianensis* may have gained a predominant advantage in order to adapt to the changing environment. In conclusion, to survive in adversity, *V. fujianensis* has enhanced its environmental adaptability and developed various strategies to fill its niche.

## 1. Introduction

The genus *Vibrio* is ubiquitous and abundant in oceanic, estuarine, and freshwater environments [1,2,3]. They are known to produce biofilm on the surface, and they either swim freely in marine water or adhere to/live associated with other organisms. More and more novel species have been scientifically identified, with more than 130 *Vibrio* species reported to date. Many *Vibrio* spp. are well-known bacterial pathogens, causing disease in humans or marine animals. *Vibrio cholerae* [4] is the causative agent of human epidemic cholera. *Vibrio parahaemolyticus* causes severe gastroenteritis in humans through consumption of contaminated seafood [5]. Some other *Vibrio* can also cause severe bacteremia, skin, and soft tissue infection [6].

In the process of evolution, *Vibrio* adapted to its environment in various strategies. Firstly, *Vibrio* can obtain a better survival ability in their environment by forming biofilm or growing rapidly to a certain population density. Many *Vibrio* isolates use population density to control the gene expression via a quorum-sensing system; for instance, the quorum-sensing transcription factor AphA directly regulates natural competence in *V. cholerae* [7]. *V. cholerae* also possesses multiple quorum-sensing systems that control virulence and biofilm formation among other traits [8]. Secondly, *Vibrio* spp. can develop adaptive strategies to survive in extreme conditions of salinity stress and temperature, such as the mechanism of osmoregulation and osmotic balance, modification of the lipid composition, activity of ion pumps, and increasing of the secondary metabolite production [9]. In addition, one of the adaptive strategies employed by various *Vibrio* spp. to survive in the changing environment is genetic variation via positional mutations or the horizontal transfer of foreign genes [10].

*Vibrio fujianensis* was recently identified as a novel *Vibrio* species; the type strain FJ201301^T^ was isolated from aquaculture water in Fujian Province, China, in 2013 [11]. As a novel species, *V. fujianensis* has evolved many characteristics and survival strategies to occupy its niche. For instance, the strain of *V. fujianensis* is able to grow under a wide range of pH (pH 5–10) and salt concentrations (1%–10% *w*/*v*) [11]. In addition, the strain of *V. fujianensis* can result in a cross-agglutination reaction with the specific serum of *V. cholerae* O139 serogroup, which suggests the two species have a similar or the same O-antigen component. It is necessary to understand the evolutionary characteristics and genomic diversity of a new *Vibrio* species. In the present study, a comparative genomics analysis was performed based on the *V. fujianensis* draft genome and other reference genomes of the genus *Vibrio*. Two different salt stress conditions, 1% (*w*/*v*) and 8% (*w*/*v*), were selected for RNA-Seq-based transcriptome analysis. We investigated the comparative differentially expressed gene (DEG) profiles of *V. fujianensis* with low-salt or high-salt stress, so as to gain an increased understanding of the molecular mechanisms underlying the species’ environmental response.

## 2. Materials and Methods

### 2.1. Strain Culture and RNA Preparation

*V. fujianensis* FJ201301^T^ was used throughout this study. The strain was cultured as recommended for Luria-Bertani (LB, 3%, NaCl, *w*/*v*) agar at 30 °C for 18 h. The total RNA extraction from the two conditionally cultured strains was performed using the TRIZOL extraction reagent (Invitrogen, Carlsbad, CA, USA), in accordance with the protocol mentioned before [12].

### 2.2. Genome and Transcriptome Sequencing

The whole genome sequence of *V. fujianensis* FJ201301^T^ (accession number: GCA_002749895.1) was sequenced previously. In order to represent low-salt stress and high-salt stress, 1% and 8% (NaCl, *w*/*v*) concentrations were chosen, respectively. The RNA samples were extracted separately in each condition to measure them three times in duplicate. Transcriptome sequencing was performed using the Illumina HiSeq™ 2000 platform (San Diego, USA), with an average yield of 10.52 Mb raw data per sample. SOAPnuke software [13] was used to remove the adapters and low-quality reads in the sequencing data. Then, the filtered reads were compared to the reference genome using HISAT 2.1.0 software [14]. The whole RNA-sequencing process, including the RNA library construction, sequencing, and data pipelining, was done in accordance with the manufacturer’s protocols by a commercial sequencing service (BGI, Shenzhen, China).

### 2.3. Phylogenetic Analysis

Phylogenetic analysis was done using the 16S rRNA gene sequences and single-copy gene sequences. 16S rRNA gene sequences of 119 *Vibrio* bacteria and *Aeromonas hydrophila* ATCC 7966^T^ are available in the GenBank (https://www.ncbi.nlm.nih.gov/genbank/; Appendix A). The genetic distance and sequence similarity of the 16S rRNA genes were calculated with MEGA (version 7.0) [15] using Kimura’s two-parameter model. A maximum likelihood phylogenetic tree was reconstructed by MEGA software with 1000 bootstrap replicates. Gene prediction was performed using the prodigal tool in the prokaryotic genome sequences. Following the definition of a single-copy core gene by Fabini et al. [16], genes with a single-copy characteristic in each strain were identified as the core genes. After gene prediction from the genomes, CD-HIT (version 4.6.6, 2016) [17] was used to calculate a nonredundant homologous gene set. Next, we compared the coding genes of each strain with this nonredundant gene set using BLAST+ [18]. The core genome and the accessory genome were investigated in order to identify shared and unique genes, and to explain differences among several *Vibrio* species. The shared and unique genes in a Venn diagram were determined using BLASTn with an E-value cutoff of 1.0 × 10^−5^. The Venn diagram was reconstructed using R-3.5.2 [19]. DNA–DNA hybridization (DDH) values were computed using the Genome-to-Genome Distance Calculator v.2.1 [20]. The tools can be accessed from the following online resource: http://ggdc.dsmz.de/. The DDH values were calculated using the second formula. Genomes of *V. fujianensis* FJ201301^T^ and other *Vibrio* species (Appendix A) were used for phylogenetic analysis.

### 2.4. O-Polysaccharide (O-PS) Gene Cluster Analysis

The O-PS gene cluster sequence of *V. fujianensis* FJ201301^T^ was aligned against that of *V. cholerae* O139 serogroup MO45. The genome region of *Vibrio cincinnatiensis* NCTC 12012^T^ and *V. metschnikovii* JCM 21189^T^ flanked by the *rfaD* and *mutM* genes was extracted from the draft genomes as the O-PS genetic region. The Rapid Annotation using Subsystem Technology (RAST; version 2.0) server pipeline [21] was used to predict the open reading frames (ORFs) and to annotate the ORFs of the O-PS gene cluster. Web-BLAST (https://blast.ncbi.nlm.nih.gov/Blast) [22] was applied to the inferred putative functions. A comparative analysis of homologous regions of the *Vibrio* O-PS gene cluster was performed by Easyfig 2.2.3 software [23].

### 2.5. Virulence Gene Analysis

The Virulence Factors Database (VFDB) [24] (http://www.mgc.ac.cn/VFs/) was used to annotate the virulence genes. The virulence-related genes of the protein sequences were searched against the VFDB using BLAST+ [18].

### 2.6. Identification of Differentially Expressed Genes

RSEM software (v1.2.8) [25] was used to calculate the gene expression level of each sample. The fragments per kilobase of transcript per million fragments mapped (FPKM) method [26] was used to describe the transcript expression. A false discovery rate (FDR) [27] of <0.001 was used as the threshold *p*-value in multiple tests to judge the degree of difference in the unigenes expression. In a given RNA-sequencing library [28], the DEGs were defined with a cutoff *p*-value ≤0.001 and a ≥2-fold-change compared between two samples.

### 2.7. Transcriptome Data Analysis

The DEGs were annotated against the Swiss-Prot [29], Gene Ontology (GO) [30], and Kyoto Encyclopedia of Genes and Genomes (KEGG) databases [31] by BLAST+ with an E-value cutoff of 1.0 × 10^−5^. The GO term functional analysis and KEGG pathway enrichment analysis were performed in subsequent analyses.

## 3. Results

### 3.1. Evolutionary Position of V. fujianensis Species

The 16S rRNA gene sequence of *V. fujianensis* FJ201301^T^ was aligned and compared to a set of 119 corresponding sequences of other *Vibrio* species and strains. *Aeromonas hydrophila* ATCC 7966^T^ served as an outgroup species. A phylogenetic tree suggested that the *Vibrio* genus should be divided into seven different clades (Figure 1A). Highly consistent with our previous study [11], the novel *V. fujianensis* FJ201301^T^ was classified as a member of the Cincinnatiensis clade; this clade also included other *Vibrio* species such as *Vibrio cincinnatiensis* NCTC 12012^T^, *Vibrio metschnikovii* JCM 21189^T^, *Vibrio bivalvicida* 605^T^, and *Vibrio salilacus* DSG-S6^T^.

A phylogenetic tree based on single-copy homology genes (Figure 1B) showed that 10 strains were divided into three main groups, group I to group III. At the genome level, *V. fujianensis* FJ201301^T^ was more closely related to *V. cincinnatiensis* NCTC 12012^T^, implying that they probably shared common ancestors in the past, according to the phylogenetic relationship. Compared with the 16S rRNA gene phylogenetic tree, the genetic relationship was slightly different. Here, *V. salilacus* DSG-S6^T^ separated from the Cincinnatiensis clade, while the remaining Cincinnatiensis clade *Vibrio* spp. still gathered closely, which were divided into group III together with the *V. cholerae* strains. The DDH value(s) indicated that interspecies distinction among *Vibrio* species in the Cincinnatiensis clade was apparent (Table 1). The DDH values among these type strains were 18.70–65.90%. These DDH values were lower than the proposed cutoff value (70%) for species delineation [32], which confirmed that they represent different members of various genomic species. Likewise, the consistent results were indicated by average nucleotide identity (ANI) and average amino acid identity (AAI) analyses (data are not shown).

### 3.2. Population Genomic Analysis

The pan-genome of the *V. fujianensis* FJ201301^T^*, V. cincinnatiensis* NCTC 12012^T^, *V. metschnikovii* JCM 21189^T^, and *V. cholerae* MO45 strains shared 817 core genes (Figure 2A), and these four genomes possessed 1301, 1409, 2115, and 2643 unique genes, respectively. An additional gene set, ranging from 178 to 1881, was shared by any two species, while the number of genes that were shared by any three species varied from 82 to 882.

The numbers of core genes were calculated from the genomes mentioned above (Figure 2B). In terms of quantity, each *Vibrio* species contained more than 3000 core genes, and *V. cholerae* MO45 had slightly more genes than the other three bacteria. Coincidentally, both *V. fujianensis* FJ201301^T^ and *V. metschnikovii* JCM 21189^T^ had 3284 core genes. On the other hand, the decreasing trend (Figure 2C) became more apparent when the number of strains continued to increase.

### 3.3. Comparative Analysis of O-PS Gene Cluster

The O-PS gene cluster analysis (Figure 3A) showed that *V. fujianensis* FJ201301^T^ had a few sequence variations compared with *V. metschnikovii* JCM 21189^T^ and *V. cincinnatiensis* NCTC 12012^T^, resulting in a single branch separately in the clustering analysis. A comparison of the O-PS genes distribution (Figure 3B) revealed the possible reason for this large variation. The O-PS gene cluster sequence of *V. fujianensis* FJ201301^T^ is partially similar to the counterparts of *V. metschnikovii* JCM 21189^T^ and *V. cincinnatiensis* NCTC 12012^T^. We found that the O-PS gene cluster of *V. fujianensis* had three homologous fragments shared with the same region of *V. cholerae* MO45 (the similarity was 98.5%, 98.0%, and 97.8%, respectively). The first two homologous regions contained four genes, namely, *manB*, *manC*, *wbfR*, and *wbfS* (Figure 3C), encoding phosphomannomutase, mannose-1-phosphate guanylyltransferase, UDP-N-acetylgalactosaminyltransferase, and asparagines synthetase, respectively. The third region contained the *gmd* gene and nine other genes, namely, *wbfH*, *wbfI*, and *wbfJ* to *wbfP* (Figure 3C). Gene *gmd* catalyzed the conversion of GDP-D-mannose to GDP-4-dehydro-6-deoxy-D-mannose. The *wbf* family genes were likely to participate in the regulation of the cell wall biosynthesis, and may be involved in the maturation of the outermost layer of protein. In addition, considering the DNA GC content, the O-PS gene cluster had a GC content (40.9 mol%) lower than the genome average (43.4 mol%), while the homologous fragment (nucleotide site from 990 to 20,158) had a GC content of 38.2 mol%, which was far lower than the average GC content of the genome (data are not shown). This atypical GC content provided strong evidence that the homologous fragment had recently gone through a genetic recombination event, by horizontal gene transfer, with *V. cholerae* O139 serogroup strains, a different bacterial species in the long-term evolution.

### 3.4. Prediction of Virulence-Associated Genes

The virulence-associated factors of *V. fujianensis* FJ201301^T^ and other common pathogens were predicted against the VFDB. A putative virulence-associated gene pool included a total of 281 kinds of virulence genes (Appendix A), and approximately 37.7% of these genes were found in the *V. fujianensis* FJ201301^T^ genome. *V. fujianensis* FJ201301^T^ carried 17 kinds of potential virulence factors and 106 related genes (Table 2). The genes involved in mannose-sensitive hemagglutinin (MSHA), type IV pilus, flagella, and EPS type II secretion system were mostly found in *V. fujianensis* FJ201301^T^ and pathogenic *Vibrio* species (Appendix A). Cholera toxin *ctxA*, thermosolve hemolysin *tlh*/*tdh*, and other toxins (e.g., *hlyA* and *rtxA*/*B*) were not detected in *V. fujianensis* FJ201301^T^. The unknown protein related to O-antigen has been described only in *V. fujianensis* FJ201301^T^ and in *V. cholerae* MO45 (Appendix A). Moreover, *V. fujianensis* FJ201301^T^ and *V. cincinnatiensis* NCTC 12012^T^ were still the most similar in the aspect of the category and quantity of virulence factors (Figure 4). Thus, we suspected that the potential pathogenicity of *V. fujianensis* FJ201301^T^ was due to multi-interaction with a variety of virulence factors.

### 3.5. Gene Expression Analysis

A comparative transcriptome analysis between the low-salt and high-salt condition was conducted so as to identify the genes involved in salt tolerance. A Pearson correlation coefficient of 0.754 was calculated to reflect the correlation of the unigenes expression between two samples. The length distribution of the transcripts is shown in Appendix A.

The box plot of the total gene expression (Figure 5A) showed the distribution and dispersion of the gene expression levels in two conditions. Comparing the relative transcript abundance in each unigene by using the FPKM, a total of 913 unigenes were found to be differentially expressed (Figure 5B); in a high-salt condition, 577 of these were significantly upregulated, whereas 336 unigenes were significantly downregulated. A cluster heatmap analysis of the DEG patterns is shown in Appendix A.

### 3.6. Annotation Analysis of DEGs

The GO annotation analysis (Figure 6A) categorized the DEGs into three modules, namely, biological process, cellular component, and molecular function. The GO terms predominantly enriched in the whole unigene pool were associated with molecular function, including “catalytic activity” for the first place, and “binding” ranked secondly (data are not shown). The top five GO terms mainly relevant for the upregulated DEGs contained “catalytic activity”, “binding”, “cellular process”, “membrane”, and “membrane part”. The downregulated genes group was largely associated with the terms “catalytic activity”, “binding”, “metabolic process”, “cellular process”, and “organic substance metabolic process” (data are not shown). The GO enriched bubble graph with the top 20 GO terms ranked by the smallest Q-value showed the different degrees of up- or downregulated DEGs enrichment from three dimensions (Appendix A). Notably, all of these DEGs significantly enriched the GO terms of “structural constituent of ribosome” and “ribosome”.

To get a further comprehensive understanding of the enriched metabolic or signal transduction pathways, we classified the DEGs in the KEGG pathways database for enrichment analysis (Figure 6B, top 20 terms ranked by the smallest Q-value). The KEGG pathway annotation classification of the DEGs between low-/high-salt environment is shown in Appendix A. As a result, “ribosome pathway” (ko03010) was of significance in the enrichment analysis (Q-value < 0.05).

### 3.7. Key Genes under Salt Tolerance Response

The ribosome is described as a place for protein biosynthesis in cells and has a considerable influence on salt tolerance. There were 32 candidate genes found in the “ribosome pathway” (ko03010) that may participate in changes in the ribosome formation, eight of which were significantly upregulated, while the remaining 24 genes were significantly downregulated (Table 3 and Figure 7). Transcriptional changes in the ribosomal genes indicated that the *V. fujianensis* species responded efficiently and quickly to high-salt stress, which might be a predominant advantage in the changing environment.

## 4. Discussion

To date, more than 130 species are recognized in the genus *Vibrio*, many of which have been identified in recent years. It has been shown that the *Vibrio* genus is characterized by a remarkable biodiversity, having evolved to develop complex lifestyles. In this study, the phylogenetic tree based on the 16S rRNA gene sequences of 119 *Vibrio* species defined seven phylogenetic clades (Figure 1A). *V. fujianensis* FJ201301^T^ was related to the Cincinnatiensis clade. In 2014, Michael et al. [33] used the multilocus sequence analysis (MLSA) method to study the phylogenetic relationship and identification of 56 species in the genus *Vibrio*. At that time, *V. cincinnatiensis* and *V. metschnikovii* were classified as Cholerae clade with robust bootstrap values. However, in recent years, a considerable quantity of novel *Vibrio* species have been continuously identified and published, such as *V. oceanisediminis* [34] and *V. injenensis* [35], which have entered and broadened our horizon. Interpretation of bootstrap values makes these newly reported species separate from the Cholerae clade based on the phylogenetic tree. We propose that the Cincinnatiensis clade is an evolutionarily independent *Vibrio* clade that consists of these *Vibrio* species. *V. fujianensis*, as a novel species, is deemed to expand the species abundance and genetic diversity of the genus *Vibrio.*

A gene family is a group of genes derived from the same ancestor and consisting of two or more copies of a gene through gene duplication [36] and species divergence [37,38] in the biosphere. In most cases, structural genes are in a single-copy form in the bacterial chromosomal genome. *V. cincinnatiensis* and *V. metschnikovii* were recognized as two closest species of *V. fujianensis*. Great differences in the representative genomes of the four *Vibrio* strains (*V. fujianensis* FJ201301^T^, *V. cincinnatiensis* NCTC 12012^T^, *V. metschnikovii* JCM 21189^T^, and *V. cholerae* O139 serogroup MO45) were found, which somehow would cause differences in some aspects, like the GC content, genome structure, proportion of core genes and accessory genes. Each *Vibrio* strain contained a large number of strain-specific genes, which may be related to different strains inhabiting a limited host and environmental niche, or related to distinct types of diseases with different severity [39]. These strain-specific genes with an unequal number and diverse functions might confer several microbes some potential to go through environmentally troubled times [40,41].

Virulence-associated genes are believed to allow strains to carry adaptation and pathogenicity [42]. Studies of the virulence factor increasingly show the important role of virulence-associated genes in environmental adaptation. Virulence-related profile, including flagella, pili, and the two-component regulatory system, indicated the putative differentiation in niche and pathogenicity [38]. Meanwhile, the virulence genes were associated with environmental stress adaptation [43]. Virulence factors in *M. tuberculosis* were implicated in the adaptation of limited nutritional conditions in macrophages or for counteracting the microbicidal host cell responses, and *M. tuberculosis EspC* and *EspA* mutants have shown an inhibition of the bacterial growth or the biological metabolism [44]. In our study, the VFDB prediction result showed that the MSHA, type IV pilus, flagella, and EPS type II secretion system-related virulence genes were detected in *V. fujianensis*, and these genes were universally shared in common pathogenic *Vibrio* species. By assembling surface antigen structure [45], secreting virulence effect protein [46], and developing other lifestyles, bacteria become so powerful in order to escape the immune defense system in the host. Likewise, *V. fujianensis* has a strong adaptability to new environments. It carries a number of virulence-associated factors in the virulence-gene pool among pathogenic *Vibrio*
Table 2 and Appendix A, indicating that there may be an extensive gene swap or gene exchange between *V. fujianensis* and other pathogenic *Vibrio*.

Horizontal gene transfer (HGT) is considered to be an important mode of acquiring new genes and accelerating evolution. Mobile genetic elements, such as plasmids, phages (prophages), and integrons, are found to be a crucial factor for horizontal gene transfer, even contributing to the virulence and antibiotic resistance of *Vibrio* [47]. Recently, the authors [48] showed that genetic material via HGT promoted the environmental adaptability of pathogenic bacteria. Borgeaud et al. [49] have confirmed that in *V. cholerae*, type VI secretion system genes are coregulated with genes involved in exogenous DNA uptake. On the other hand, cases of serum cross-agglutination between different species are reported from time to time; a well-known example is the cross-reaction between *E. coli* and *Shigella* in *Enterobacter* species [50,51,52]. In our research, we found that the O-PS gene cluster of *V. fujianensis* FJ201301^T^ and *V. cholerae* O139 serogroup MO45 shared highly homologous gene coding regions. This finding was a reasonable explanation for the cross-agglutination between *V. fujianensis* and *V. cholerae* O139 serum. With combined atypical GC content values, we propose that between the *V. fujianensis* FJ201301^T^ and *V. cholerae* O139 serogroup MO45, a horizontal gene transfer of a partial O-PS gene cluster may have occurred. Sally et al. [53] demonstrated the adaptive role of modifications to the LPS structure (including loss of O-antigen expression) in *Pseudomonas aeruginosa* over the course of chronic respiratory tract infections. The O-antigen of *V. cholerae* is proven to play a role in critical defense by increasing drag force to impede attackers [54]. O-antigen is also the receptor of the *V. cholerae* specific typing phage [55]. Another research showed that the O-antigen in *Vibrio* species was associated with the specific environmental colonization ability [56], which can be beneficial to develop alternative lifestyles during intestinal colonization. Therefore, we consider that the exchange of partial O-antigen fragments between *V. fujianensis* FJ201301^T^ and *V. cholerae* O139 can greatly enhance the environmental adaptability of *V. fujianensis* FJ201301^T^ to quickly reproduce and tenaciously survive in nature.

Transcriptome changes for *V. fujianensis* in response to salt stress tolerance are mainly reflected in the molecular function represented by catalytic activity. Transcriptional changes involved in the carbohydrate metabolism, membrane transport, amino acid metabolism, and signal transduction were found to be important for the high salt stress in *V. fujianensis*. The KEGG pathway enrichment analysis considered the “ribosome pathway” (ko03010) as a significant enrichment; this result is consistent with the high-salt transcription and expression of *Shewanella algae* in our previous study [57]. Eiji et al. [58] found that the VemP-mediated regulation of SecDF2 was essential for the survival of marine bacteria when the Na^+^ concentration in the environment changed. Transcriptome research in response to salt stress in *Betula halophila* [59] showed that the top four enriched pathways were “fatty acid elongation”, “ribosome”, “sphingolipid metabolism”, and “flavonoid biosynthesis”; furthermore, the related transcription factor was analyzed by qRT-PCR in order to verify the important role in salt resistance in *B. halophila*. Proteomic analyses in *C. albicans* revealed a link between ribosomal gene expression and environmental adaptation [60]. Salt tolerance is a common feature of *Escherichia coli*, in which the maturation or function of the ribosome is impaired [61]. Changes in the ribosome confer salt resistance on some microbes [57,61]. Microbes begin to synthesize or uptake osmo-protectants in a high-salt condition, while inhibiting general σ^70^ transcription [62]. The Na^+^/H^+^ antiporter makes it effective for the survival of *Vibrio* species in a saline environment. *NhaB* has been proven to be a possible mechanism for regulating Na^+^/H^+^ antiporter activity [63]. To our knowledge, some microorganisms form a variety of biological mechanisms and physiological responses to high salt stress, and those mechanisms can help them take advantage of the unfavorable environments [9,64]. Marsden et al. [65] found that nutrient and salt availability may contribute to *Vibrio* biofilm formation. In general, microorganisms rely on a salt rejection strategy called osmoadaptation, which involves compatible solute accumulation, for example, the accumulation of K+ ions or some low molecular mass organic solutes. Usually, the formation of organic solutes makes conditions more favorable for the osmotic balance in cells, like sugar, alcohol, amino acid, and their derivatives [66]. In addition, mediation by the Na^+^/H^+^ antiporter is used to pump extra Na^+^ ions out of the cell to maintain a proper Na^+^ concentration in the cell. Fu et al. [57] also found that genes involved in peptidoglycan synthesis, DNA repair, tricarboxylic acid cycle, and the glycolytic pathway were the alternative emphasis of salt response mechanisms in *Shewanella algae*.

## 5. Conclusions

In this study, changes in the ribosome pathway may be related to salt tolerance via transcriptome sequencing. Strain-specific genes in the *V. fujianensis* species might help develop its innate adaptability, as well as its potential ability to evolve quickly and thrive in the ecological niche. Several genetic motility genes might be attributed to HGT in the evolutionary process.

## Figures and Tables

**Figure 1 microorganisms-08-00555-f001:**
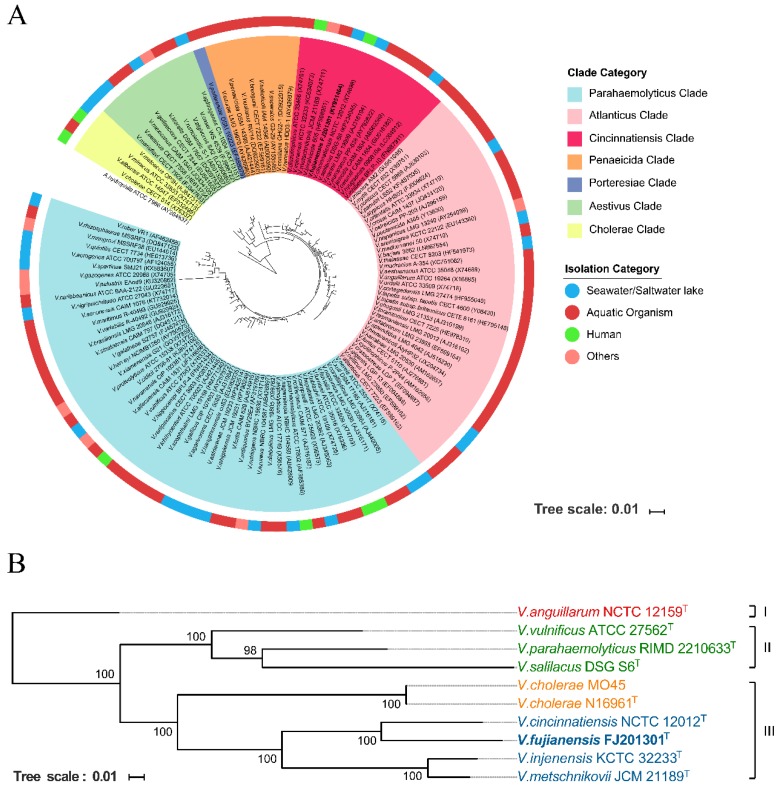
Phylogenetic analysis of *Vibrio fujianensis*. (**A**) Phylogenetic tree among the genus *Vibrio* based on 16S rRNA gene sequences. (**B**) Phylogenetic tree based on homologous gene sequences of *Vibrio* species analyzed in this study. All single-copy homologous genes for each species were concatenated to form a new sequence 97,365 bp in length. The horizonal bar represents 0.01 substitution per nucleotide site. The accession numbers of 16S rRNA gene sequences and genomes are shown in Appendix A, respectively.

**Figure 2 microorganisms-08-00555-f002:**
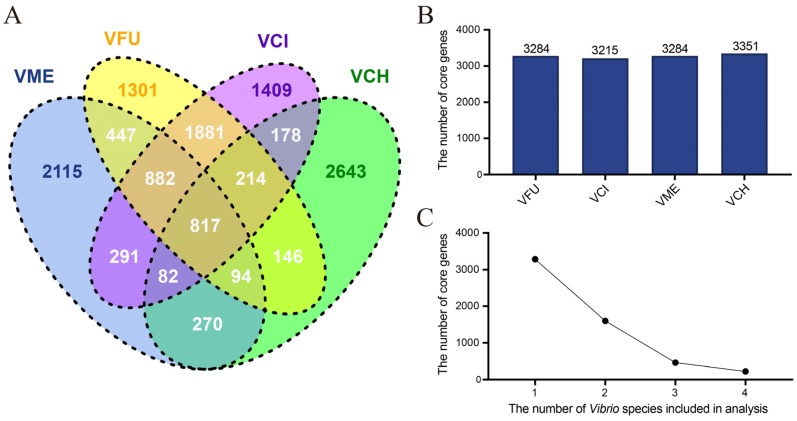
Comparative genomic analysis of *Vibrio fujianensis* and three other *Vibrio* species. (**A**) Venn diagram of the shared and unique genes found in *V. fujianensis* FJ201301^T^ and three other *Vibrio* genomes. VME: *V. metschnikovii* JCM 21189^T^; VFU: *V. fujianensis* FJ201301^T^; VCI: *V. cincinnatiensis* NCTC 12012^T^; VCH: *V. cholerae* O139 serogroup MO45. (**B**) The number of core genes shared in *V. fujianensis* FJ201301^T^ and three other *Vibrio* genomes. (**C**) Core gene quantitative trend. *Vibrio* species were added one by one for analysis in the following order (from 1 to 4): *V. fujianensis* FJ201301^T^, *V. cincinnatiensis* NCTC 12012^T^, *V. metschnikovii* JCM 21189^T^, and *V. cholerae* O139 serogroup MO45.

**Figure 3 microorganisms-08-00555-f003:**
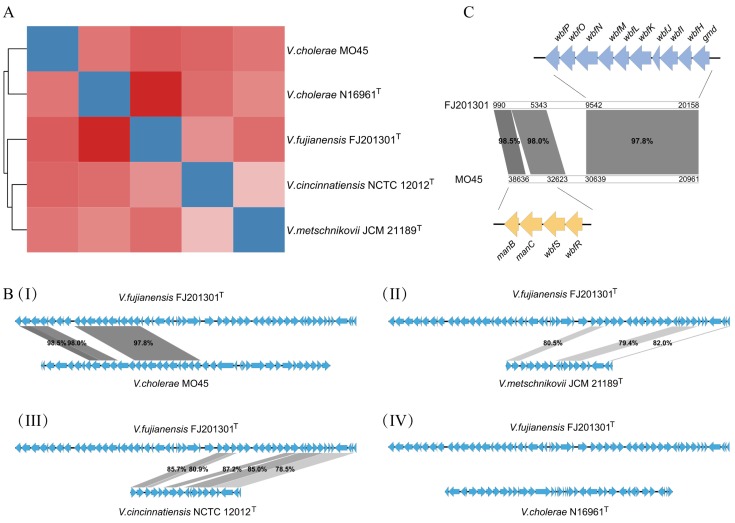
Comparative analysis of O-polysaccharide (O-PS) gene cluster. (**A**) Clustering of O-PS gene cluster. (**B**) O-PS gene cluster comparison between *V. fujianensis* FJ201301^T^ and other *Vibrio* species. I to IV show *V. fujianensis* vs. *V. cholerae* O139 serogroup MO45, *V. metschnikovii* JCM 21189^T^, *V. cincinnatiensis* NCTC 12012^T^, and *V. cholerae* O1 serogroup N16961^T^, respectively. (**C**) Homologous regions (nucleotide site from 990 to 20,158) of O-PS gene cluster between *V. fujianensis* FJ201301^T^ and *V. cholerae* O139 serogroup MO45.

**Figure 4 microorganisms-08-00555-f004:**
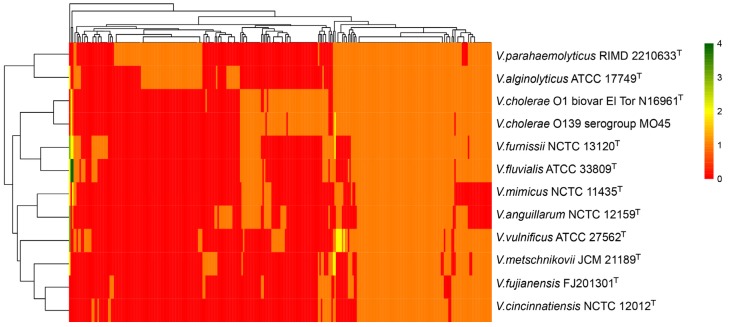
Two-dimensional hierarchical clustering analysis of putative virulence-associated genes based on the Virulence Factors Database (VFDB). Pathogenic *Vibrio* species are shown in the column and virulence factors are shown in the row. Different colors represent the corresponding number of virulence factors.

**Figure 5 microorganisms-08-00555-f005:**
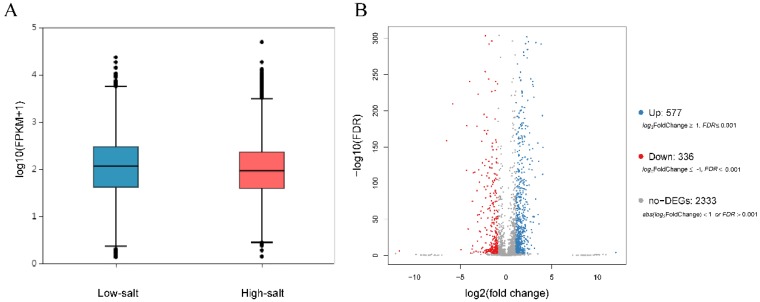
Gene expression analysis. (**A**) Box plot of the total expressed genes evaluated by FPKM method in the low-salt or high-salt condition. (**B**) Volcano plot of the genes differentially expressed between the two samples. The blue and red colors represent upregulated and downregulated genes, respectively.

**Figure 6 microorganisms-08-00555-f006:**
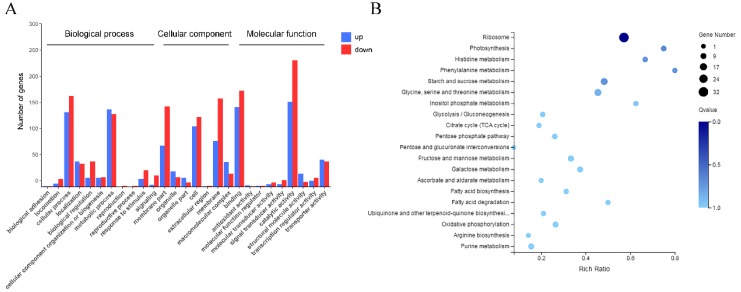
(**A**) Gene ontology (GO) functional annotation analysis of the differentially expressed genes (DEGs). (**B**) KEGG pathway enrichment analysis of DEGs between low-/high-salt condition. The bubble size indicates the number of genes, and the color shade represents the Q-value.

**Figure 7 microorganisms-08-00555-f007:**
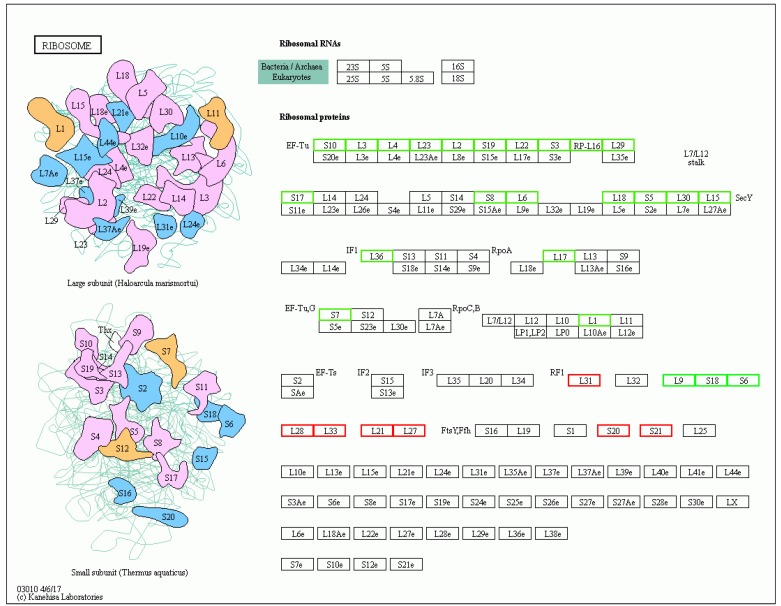
Candidate unigenes related to salt tolerance. Red frames represent upregulated genes, while green frames represent downregulated genes.

**Table 1 microorganisms-08-00555-t001:** DNA–DNA hybridization (DDH) among *Vibrio* species of the Cincinnatiensis clade.

*Vibrio* Species *	*Vibrio* Species *											
	A	B	C	D	E	F	G	H	I	J	K	L
A	100	20.30	19.50	29.60	18.90	22.20	19.00	21.70	19.50	21.70	20.20	20.30
B		100	19.40	19.80	26.50	21.50	21.50	20.80	21.50	21.30	19.10	19.30
C			100	19.90	18.70	21.10	18.90	20.50	20.80	20.60	19.20	19.00
D				100	19.40	22.00	18.90	21.80	19.60	22.10	20.20	20.50
E					100	20.60	21.10	20.10	22.00	20.10	19.10	20.00
F						100	20.90	54.00	22.30	35.30	21.00	21.50
G							100	20.20	45.70	20.90	18.70	19.00
H								100	20.80	38.70	20.80	21.20
I									100	20.60	19.10	20.20
J										100	20.60	21.30
K											100	65.90
L												100

*Vibrio* species *: A. *Vibrio bivalvicida* 605^T^; B. *Vibrio cincinnatiensis* NCTC 12012^T^; C. *Vibrio diazotrophicus* NBRC 103148^T^; D. *Vibrio europaeus* PP-638^T^; E. *Vibrio fujianensis* FJ201301^T^; F. *Vibrio hyugaensis* 090810a^T^; G. *Vibrio injenensis* KCTC 32233^T^; H. *Vibrio jasicida* CECT 7692^T^; I. *Vibrio metschnikovii* JCM 21189^T^; J. *Vibrio owensii* CAIM 1854^T^; K. *Vibrio pacinii* DSM 19139^T^; L. *Vibrio salilacus* DSG S6^T^.

**Table 2 microorganisms-08-00555-t002:** Function and pathogenic role of the virulence factors of *V. fujianensis* FJ201301^T^.

VF Class	Virulence Factors	Function and/or Pathogenic Role
Adherence	Accessory colonization factor	Signal transduction
	Mannose-sensitive hemagglutinin (MSHA type IV pilus)	Pilus assembly and pathogenesis
	Type IV pilus	Motility, cell–cell adhesion, and pathogenesis
	LPS O-antigen	Undetermined
	The tad locus	Hydrolase and tRNA processing
Antiphagocytosis	Capsular polysaccharide	LPS biosynthesis and metabolism
Chemotaxis and motility	Flagella	Flagellum biogenesis, motor activity, and pathogenesis
Iron uptake	Enterobactin receptors	Iron transport
	ABC transport systems	ATPase activity and transport
	Vibriobactin biosynthesis	Catalytic activity and multifunctional enzyme
	Acinetobactin	Enzyme activity
Quorum sensing	Autoinducer-2	Lyase, autoinducer synthesis, and quorum sensing
Secretion system	EPS type II secretion system	Protein secretion and protein transport
Others	O-antigen	Undetermined
Endotoxin	LOS	Multifunctional enzyme and lipopolysaccharide biosynthesis
Invasion	Flagella	Hydrolase and chemotaxis
Regulation	Two-component system	Transcription regulation

**Table 3 microorganisms-08-00555-t003:** Differentially expressed candidate genes involved in the ribosome pathway.

Gene ID	Gene Alias	Substates	Function
**Upregulated**			
B7C60_RS02450	*rpmE*	large subunit ribosomal protein L31	Translation
B7C60_RS04755	*rpsT*	small subunit ribosomal protein S20	Translation
B7C60_RS05710	*rpmG*	large subunit ribosomal protein L33	Translation
B7C60_RS05715	*rpmB*	large subunit ribosomal protein L28	Translation
B7C60_RS11045	*rplU*	large subunit ribosomal protein L21	Translation
B7C60_RS11050	*rpmA*	large subunit ribosomal protein L27	Translation
B7C60_RS11455	*rpsU*	small subunit ribosomal protein S21	Translation
B7C60_RS12925	*rpmE*	large subunit ribosomal protein L31	Translation
**Downregulated**			
B7C60_RS00940	*rplI*	large subunit ribosomal protein L9	Translation
B7C60_RS00945	*rpsR*	small subunit ribosomal protein S18	Translation
B7C60_RS00950	*rpsF*	small subunit ribosomal protein S6	Translation
B7C60_RS01530	*rplA*	large subunit ribosomal protein L1	Translation
B7C60_RS01740	*rpsG*	small subunit ribosomal protein S7	Translation
B7C60_RS03490	*rplQ*	large subunit ribosomal protein L17	Translation
B7C60_RS03515	*rpmJ*	large subunit ribosomal protein L36	Translation
B7C60_RS03525	*rplO*	large subunit ribosomal protein L15	Translation
B7C60_RS03530	*rpmD*	large subunit ribosomal protein L30	Translation
B7C60_RS03535	*rpsE*	small subunit ribosomal protein S5	Translation
B7C60_RS03540	*rplR*	large subunit ribosomal protein L18	Translation
B7C60_RS03545	*rplF*	large subunit ribosomal protein L6	Translation
B7C60_RS03550	*rpsH*	small subunit ribosomal protein S8	Translation
B7C60_RS03575	*rpsQ*	small subunit ribosomal protein S17	Translation
B7C60_RS03580	*rpmC*	large subunit ribosomal protein L29	Translation
B7C60_RS03585	*rplP*	large subunit ribosomal protein L16	Translation
B7C60_RS03590	*rpsC*	small subunit ribosomal protein S3	Translation
B7C60_RS03595	*rplV*	large subunit ribosomal protein L22	Translation
B7C60_RS03600	*rpsS*	small subunit ribosomal protein S19	Translation
B7C60_RS03605	*rplB*	large subunit ribosomal protein L2	Translation
B7C60_RS03610	*rplW*	large subunit ribosomal protein L23	Translation
B7C60_RS03615	*rplD*	large subunit ribosomal protein L4	Translation
B7C60_RS03620	*rplC*	large subunit ribosomal protein L3	Translation
B7C60_RS03625	*rpsJ*	small subunit ribosomal protein S10	Translation

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
