# Peer review of "Comparative Genomics and Transcriptomics Analyses Reveal a Unique Environmental Adaptability of *Vibrio fujianensis"

_microorganisms, 2020, doi:10.3390/microorganisms8040555_

Round 1
Reviewer 1 Report
This is a very interesting, well written paper that relates some novel findings about Vibrio fujianensis, focus of the authors' research. This vibrio was found to contain 817 vibrio core genes and 106 virulence genes common to other pathogenic vibrios, e.g., CtxA. In addition, they found that over 1,000 genes were either up-regulated or down-regulated, as a function salt concentration.
Specific comments
The authors need to explain the meaning of the last sentence in the abstract
Author Response
Reviewer #1 (Comments for the Author):
This is a very interesting, well written paper that relates some novel findings about Vibrio fujianensis, focus of the authors' research. This vibrio was found to contain 817 vibrio core genes and 106 virulence genes common to other pathogenic vibrios, e.g., CtxA. In addition, they found that over 1,000 genes were either up-regulated or down-regulated, as a function salt concentration.
Specific comments
The authors need to explain the meaning of the last sentence in the abstract
Response:
This conclusion is not rigorous. This sentence has been corrected as “In conclusion, to survive in adversity, V. fujianensis has enhanced its environmental adaptability and developed various strategies to fill its niche.”
Reviewer 2 Report
Summary
The novel species Vibrio fujianensis (type strain FJ201301), classified as a member of the Cincinnatiensis clade, has shown distinct similarities to the V. cholerae strain cluster following single-copy gene homology analysis. The identification of three homologous fragments to with homology to V. cholerae presented atypical GC content which could strongly suggest that recombination had occurred via horizontal gene transfer. Virulence-associated genes identified in V. fujianensis were identified however the cholera toxin was not present; this presents that the pathogenicity of this strain is likely to be due to a combination of virulence factors. The investigation of key genes involved in salt tolerance identified that the ribosome was heavily involved in high-salt tolerance which could imply that this species has adapted to a changing environment in order to survive.
Lines 34 – 36 maybe some translation issues but the first few sentences do not seem to read very well. I am not sure what they mean by water column either?
Line 46 formatting issue with quorum-sensing
Line 49 reference missing, it just says et al.
Line 69 reference – it says in accordance to protocol mentioned before but cannot see where this has been talked about…
Line 71 should this say We?
Line 77 they haven’t said how the reads where filtered e.g. parameters, software used etc?
Line 79 They haven’t said what was used to do the gene-prediction e.g. software
Line 93 they need to add VFDB in brackets
Line 109 Should they show how the 119 sequences were split between Vibrio and A. hydrophila? Or have I read that incorrectly?
Line 110 they haven’t mentioned A. hydrophila in the methods – does this need to be added?
Line 123 should this be 16S rRNA gene analysis?
Line 153 where they have put the strain number I think it would be better putting the strain name as previously and after to keep it consistent.
Figure 3A this is quite confusing and is not any clearer reading the legend
Line 237 says plant for protein – not sure what this means!
Line 26768 sentence doesn’t make sense!
Author Response
Reviewer #2 (Comments for the Author):
The novel species Vibrio fujianensis (type strain FJ201301), classified as a member of the Cincinnatiensis clade, has shown distinct similarities to the V. cholerae strain cluster following single-copy gene homology analysis. The identification of three homologous fragments to with homology to V. cholerae presented atypical GC content which could strongly suggest that recombination had occurred via horizontal gene transfer. Virulence-associated genes identified in V. fujianensis were identified however the cholera toxin was not present; this presents that the pathogenicity of this strain is likely to be due to a combination of virulence factors. The investigation of key genes involved in salt tolerance identified that the ribosome was heavily involved in high-salt tolerance which could imply that this species has adapted to a changing environment in order to survive.
- Lines 34 – 36: maybe some translation issues but the first few sentences do not seem to read very well. I am not sure what they mean by water column either?
Response:
Thanks very much for reviewer’s comment. The first few sentences mean that most bacteria live in nature in two states, one is planktonic and the other is sticky. Numbers of bacteria are swimming in the water column, and the word ‘water column’ refers to the water environment in a planktonic state. Word ‘water column’ has been deleted and changed to ‘swim freely in marine water’. (Track Changes of Final Showing Markup (or All Markup) are shown in the new manuscript Line 40.)
- Line 46: formatting issue with quorum-sensing
Response:
We have changed the correct text font and size (see new manuscript Line 52).
- Line 49: reference missing, it just says et al.
Response:
Thanks very much for reviewer’s comment. The Line 49 here lists several environmental adaptation strategies for Vibrio. Reference have also been cited. The word ‘et al.’ has been deleted to avoid misunderstanding. (see new manuscript Lines 53-59)
- Line 69: reference – it says in accordance to protocol mentioned before but cannot see where this has been talked about…
Response:
Thanks very much for reviewer’s comment. TRIZOL extraction is a method, and main reagent is an RNA-extracted Mini kit, equal to RNAiso reagent. Detailed steps were cited in the referenced article written by Fu et al.
- Line 71: should this say We?
Response:
Thanks very much for reviewer’s suggestion. The word ‘We’ should not appear in the text and has been deleted. (see new manuscript Line 84)
- Line 77: they haven’t said how the reads where filtered e.g. parameters, software used etc?
Response:
To filter raw data, we use the SOAPnuke software self-developed by BGI for removing the adapters and removing those low-quality reads in the sequencing data. Reference titled ‘The Sanger FASTQ file format for sequences with quality scores, and the Solexa/Illumina FASTQ variants.’ by Cock P., et al. has been cited. (see new manuscript Lines 92-93)
- Line 79: They haven’t said what was used to do the gene-prediction e.g. software
Response:
To do the gene-prediction, we used the prodigal tool in prokaryotic genome sequences and wrote fasta files using R package seqinr tool. (see new manuscript Lines 104-107)
- Line 93: they need to add VFDB in brackets
Response:
Thanks very much for reviewer’s attention. We had added ‘VFDB’ in brackets (see new manuscript Line 132).
- Line 109: Should they show how the 119 sequences were split between Vibrio and A. hydrophila? Or have I read that incorrectly?
Response:
Thanks very much for reviewer’s comment. Strain Aeromonas hydrophila served as an outgroup. The basis of A. hydrophila outgroups is based on a published reference written by Fang et al. (Vibrio fujianensis sp. nov., isolated from aquaculture water. Int. J. Syst. Evol. Microbiol. 2018, 68, 1146-1152).
- Line 110: they haven’t mentioned A. hydrophila in the methods – does this need to be added?
Response:
Thanks very much for reviewer’s comment. Here strain Aeromonas hydrophila ATCC 7966T served as an outgroup. We have added ‘A. hydrophila’ in the ‘Materials and Methods’. (see new manuscript Line 100)
- Line 123: should this be 16S rRNA gene analysis?
Response:
Thanks very much for reviewer’s comment. To avoid ambiguity, we have changed this expression to ‘16S rRNA gene sequence analysis’. (see new manuscript Lines 99-100, 154)
- Line 153: where they have put the strain number I think it would be better putting the strain name as previously and after to keep it consistent.
Response:
Thanks very much for pointing out the inappropriateness. We have unified the format of strain name and strain number to keep it consistent. Word ‘Strain V. fujianensis FJ201301T’ has been changed to ‘V. fujianensis FJ201301T’. Correction has been made in the new manuscript (Lines 79, 157, 163)
- Figure 3A: this is quite confusing and is not any clearer reading the legend
Response:
Thanks very much for reviewer’s comment. Figure 3A is a clustering graph based on O-PS sequences, and unnecessary legend has been deleted. (see new manuscript Fig 3A)
- Line 237: says plant for protein – not sure what this means!
Response:
Word ‘plant for protein’ means protein processing factory, plant does not refer to trees and flowers. This may be a misunderstanding caused by polysemy in English. This word has been deleted in the new manuscript Line 311.
- Line 267-68: sentence doesn’t make sense!
Response:
Thanks very much for reviewer’s suggestion. This sentence has been corrected as ‘following the definition of a single-copy core gene by Fabini et al., V. cincinnatiensis and V. metschnikovii were recognized as two member of V. fujianensis’s close relatives’ We have modified some words and sentences to make it clear. Correction has been made in the new manuscript Lines 105-107, 342-343.
Reviewer 3 Report
The manuscript titled “Comparative genomics and transcriptomics analysis reveal a unique environmental adaptability of Vibrio fujianensis by Zhenzhou Huang, Keyi Yu, Yujie Fang, Hang Dai, Hongyan Cai, Zhenpeng Li, Biao Kan, Qiang Wei, Duochun Wang contains information about the genome sequencing of V. fujianensis FJ201301T and comparative genome analysis of Vibrio species. Authors also performed transcriptome sequencing to determine transcripts related with salt tolerance.
Major comments:
- The manuscript must be reviewed by an English editor. The manuscript is really hard to read and interpret.
- Methods must be reviewed extensively because not enough information is provided to understand the results.
- Please review the methods and results sections. The results section contains information about methods, cut-off values, and software that is not listed in methods.
- There are several programs, software, and database that are not properly cited.
- The conclusion of the main manuscript is not understandable because of the writing style.
- It is not clear how authors defined phylogenetic clades.
- It is not clear how samples were used for transcriptome sequencing. More information should be provided about this procedure. There is no information in the main manuscript about the number of replicates, the number of reads by replicate, how the data was curated, and how reads were filtered.
- I encourage authors to perform Average Nucleotide Analyses (ANI), Average Aminoacid Analyses (AAI), and in silico DNA-DNA hybridization analyses to obtain metrics about genome sequence identity. These analyses and metrics are important when performing comparative genomics, and also for genotaxonomy circumscription.
Minor comments
Here I provide a list of suggestions for improving the manuscript. However the manuscript must be reviewed by an English editor before publication.
Lines 14, 33, 36, 39, 41, 42, 81, 114, 120, 174, 196: Please change the genus name “Vibrios” to Vibrio. Vibrios is incorrect!.
Line 22: aberrantly expressed. Please change the word aberrantly.
Lines 24-25: “Notably, all differentially expressed genes significantly enriched in ‘structural constituent of ribosome’ and ‘ribosome’, which had considerable influence on salt press tolerance.”. Please re-write this sentence for clarity.
Line 26-27: “Transcriptional changes in ribosomal genes mean that V. fujianensis species may be a predominant advantage in a changing environment”. Please re-write this sentence for clarity.
Line 28: “has experienced extremely genome exchange events”. Please correct this phrase.
Line 33: “The genus Vibrios are ubiquitous and abundant”. -Vibrio-. Please correct this phrase.
Line 35-36: “novel species scientifically recognized”. The -scientifically recognized- is not clear.
Line 38: V. parahaemolyticus. Please include the full scientific name the first time you name a different species.
Lines 47-49: “Secondly, Vibrios develop the adaptation strategies to survive extremes of salinity and temperature, through mechanisms of osmotic balance in their cytoplasm, modification of the lipid composition, activity of ion pumps and increase of the secondary metabolite production, et al. [9].”. Please re-write this phrase.
Line 52: “V. fujianensis”. Please include the full scientific name the first time you name a different species.
Line 59-60: “In the present study, V. fujianensis as a representative species, comparative genomics was performed based on draft genomes”. Please re-write this phrase.
Line 60: “salt press conditions”. Please correct to clarify.
Line 66: “Strain V. fujianensis FJ201301T”. Please change to V. fujianensis FJ201301T or V. fujianensis strain FJ201301T in the whole manuscript.
Lines 71-73: “We had sequenced the whole genome sequence of V. fujianensis FJ201301T (Accession Number: GCA_002749895.1), and other 16S rRNA reference sequences (Table S1) closely related to taxa of the genus Vibrio were available in the GenBank database”. This phrase needs to be re-writted.
Lines 73-74: “1% and 8% (NaCl, w/v) concentrations were chosen to represent low-salt stress and high-salt stress”. Please re-write for clarity.
Lines 76-77: “with an average yield of 10.52 Mb raw data per sample, then filtered reads were compared to the reference genome by HISAT 2.1.0 software”. Please include more information about the samples, and about how you filtered and processed reads. Please include all references used.
Line 79: “After a gene-prediction in the coding sequences”. Please correct
Line 81: “Strain V. fujianensis FJ201301T”. Please change.
Line 82: “included into evolution analysis of single-copy homologous genes”. Please re-write for clarity.
Line 82-83: “Genes with a single-copy characteristic in each strain was identified as core genes”. This definition is not clear, please re-write.
Line 83-84: “Maximum likelihood phylogenetic tree was reconstructed by MEGA (version 7.0) software”. Please include the reference for MEGA, and further explain how you perform this analyses. For instance, what was the selected substation model? How the tree is supported?, no branch-support values are shown.
Line 87: “V. cincinnatiensis”. Please include the complete scientific name.
Line 91: “Web-BLAST (https://blast.ncbi.nlm.nih.gov/Blast) was applied to inferred”. Please include a reference for Web-BLAST.
Line 98: “FPKM (Fragments Per Kilobase of transcript per Million fragments mapped)”. I would suggest changing the order: first include the full name and then the abbreviation on parenthesis.
Line 99: “to calculate the transcript expression levels”. Please provide further information in the method section about this procedure.
Line 99: “FDR (false discovery rate)”. I would suggest changing the order: first include the full name and then the abbreviation on parenthesis.
Line 95: “VFDB”. I would suggest to include the full name for this abbreviation for clarity.
Lines 95-96: “To obtain high confidence of virulence related genes, VFDB blast cutoff value was set as the
96 identity 90% with coverage cutoff 80%, E-value of 1e-5.”. Please re-write this phrase for clarity.
Line 97: “(DEGs)”. I would suggest to avoid including an abbreviation in a subtitle, and better include it in the text. Line 101: DEGs were defined. Here you can include the complete name and the the abbreviation.
Lines 104-105: “The DEGs were annotated against the Swiss-Prot, GO, and KEGG databases by BLAST+ with a cut-off E-value of 10-5”. Please re-write this phrase for clarity.
Line 109: “The 16S rRNA gene sequences was determined”. Please correct “sequences was”.
Lines 109-110: “those 119 sequences of the genus Vibrio bacteria and strain Aeromonas hydrophila ATCC 7966T”. Please specify which 119 sequences.
Lines 111-112: “Vibrio genus can be approximately divided into seven different clades as the phylogenetic tree demonstrated (Figure 1A)”. You need to re-write this phrase for clarity.
Line 114: V. metschnikovii, V. bivalvicida, V. salilacus. Please include full scientific names.
Lines 116-117: “Single-copy homology based phylogenetic tree (Figure 1B) showed that 10 strains were apparently divided into three main groups, group I to group III, respectively”. Please re-write this phrase.
Line 118: “strain V. fujianensis FJ201301T”. Please write the scientific name properly.
Lines 118-119: “had the closest evolutionary relationship with V. cincinnatiensis NCTC 12012T, implying these two species may occupy the similar functional ecological niche”. Please further explain this sentence. It is not clear how you can infer that if these two species are closely related they may have the same functional ecological niche. This must be reviewed.
Lines 119-121: “Compared with the 16S rRNA gene phylogenetic tree, except for V. salilacus DSG-S6T, the remaining Vibrios of the Cincinnatiensis clade still formed a distinct cluster relatively similar to V. cholerae strains cluster”. Please re-write this phrase.
Line 123: V. fujianensis. Please include the full scientific name.
Lines 123-124: “Phylogenetic tree among the genus Vibrio based on 16S rRNA gene ---sequences---. “(B) Phylogenetic tree of the single-copy homology”. This figure title is not clear.
Line 127: “The accession number of 16S rRNA gene and genomes were shown”. Please correct number(s) gene (sequences).
Lines 132-134: “An additional gene set ranged from 178 to 1881 were shared by any two species, while any three species had co-ownership of range from 82 to 882 genes together”. Please re-write for clarity. What do you mean by “co-ownership of range”. Please correct.
Lines 136-137: “In terms of quantity, each species universally contained more than 3,000 core genes, and
137 V. cholerae MO45 was slightly higher than that of the other three Vibrios”. Please re-write this sentence.
Line 130: V. metschnikovii. Please include the full scientific name.
Line 135: “Comparison of the number of core genes were calculated among genomes”. Please correct this sentence.
Lines 138-140: “On the other hand, the decreasing trend (Figure 2C) had become more apparent as the number of Vibrio species included in the analysis increases”. Please re-write this sentence.
Line 142: “Comparative genomic analysis of V. fujianensis”. The figure includes information for several Vibrio strains not only V. fujianensis. Please correct.
Lines 143-144: “The shared and unique genome among compared genomes”. Please re-write for clarity.
Lines 148-149: “(C) Core genes dilution curves in the genomes”. Please re-write this sentence. It is not understandable.
Line 153: “strain FJ201301T”. Please include the full scientific name.
Line 160: Mannose. Change to mannose.
Line 161: Asparagine. Change to asparagines.
Lines 180-181: “We had predicted the virulence-associated factors of the V. fujianensis FJ201301T genome and common pathogens against virulence factor database (web-VFDB).”. Please re.write this sentence and provide a reference for the database.
182: “kinds of virulence genes”. How do you define a “kind” of virulence gene?.
Table 1: V. fujianensis. Please include the full scientific name.
Line 196: VFDB. Please include the full name of the abbreviation for clarity.
Line 198: “R 3.5.2 software”. Please correct and properly cite this software.
Line 201: “genes involved in two salt tolerance”. Please correct.
Lines 202-203: “The length distribution of the transcriptome”. Please re-write this phrase. It is not clear what do you mean by “length distribution of the transcriptome”.
Line 204: The box plot. Please provide further information. A box plot of what?
Line 206: “aberrantly expressed”. Please change aberrantly.
Lines 211-213: “Transcriptome gene expression analysis. (A) box plot of total gene expression. (B) volcano plot of gene expression. The blue and red color indicate up-regulated and down-regulated genes, respectively”. Please re-write this phrase.
Lines 237-238: “The ribosome, one of the most important organelles, is described as a plant for protein biosynthesis in cells and has had considerable influence on salt press tolerance”. Please re-write the phrase. The “one of the most important organelles” can be omitted. “described as a plant for protein biosynthesis in cells” what do you mean by plant, and please provide a reference to support this phrase.
Lines 241-243: “Transcriptional changes in ribosomal genes mean that V. fujianensis species respond more efficiently and quickly to high-salt press, which may be a predominant advantage in a changing environment”. Please re-write for clarity and please provide further information to support this claim.
Lines 247-248: “Candidate unigenes related to salt press. Red represents up-regulated, while green represents down-regulated”. Please re-write and provide further information to interpret the figure.
Lines 267-268: “We found that great different in the representative genomes”. Please correct this sentence.
Lines 269-271: “which had significant influence on proportion of core genes and accessory gene, consequencely produced diverse environmently-adaptive phenotypes depending on the genotype [26]”. Please re-write for clarity.
Lines 274-275: “might confer body capability of being a strong and secure super organism through environmental-troubled times”. This phrase is out of context, please re-write or further explain it.
Lines 331-332: “To our knowledge, microorganisms form a variety of physiological mechanisms and invoke biological strategies that adapt to high salt environment”. Please re-write and provide references for this sentence.
Lines 333-334: “On the one hand, they rely on a salt rejection strategy for compatible solute accumulation, which usually contains polarizable chemical substances. Please re-write for clarity and please explain what do you mean by polarizable chemical substances.
Lines 286-288: “By assembling surface antigen structure [33] of bacteria, secretion of virulence effect protein [34], and other ways to arm itself, so as to escape the immune defense system in host, V. fujianensis has strong adaptability to a new environment”. Please re-write.
Lines 289-292: “It carries a considerable number of virulence-associated factors in pathogenic Vibrios virulence-gene pool (Table 1, Table S3), indicating that there are extensive gene communication and gene transfer between V. fujianensis and other Vibrios in the water environment, and one bacterium donates genetic material to another”. Please re-write for clarity.
Line 301: we find. Please change to we found.
Line 302: “This finding well explains the phenomenon”. Please correct the sentence.
Lines 326-327: “The decrease in ribosome biogenesis and translation in C. albicans [48] by proteomic analyses was also statistically significant”. This sentence is out of context please provide more information or connect it better to the context of the manuscript.
Lines 327-328: “The discovery of the high-resolution structure of the ribosome, especially the key active sites, is decisive for salt tolerance”. Please re-write for clarity and provide further information, support and references for this claim.
Lines 328-330: “Na+/H+ antiporter of Vibrio has effect on survival in salt stress environment, and NhaB operates is proved to be a possible mechanism of regulating Na+/H+ antiporter activity [49]”. Please re-write for clarity.
Author Response
Reviewer #3 (Comments for the Author):
The manuscript titled “Comparative genomics and transcriptomics analysis reveal a unique environmental adaptability of Vibrio fujianensis by Zhenzhou Huang, Keyi Yu, Yujie Fang, Hang Dai, Hongyan Cai, Zhenpeng Li, Biao Kan, Qiang Wei, Duochun Wang contains information about the genome sequencing of V. fujianensis FJ201301T and comparative genome analysis of Vibrio species. Authors also performed transcriptome sequencing to determine transcripts related with salt tolerance.
Major comments:
- The manuscript must be reviewed by an English editor. The manuscript is really hard to read and interpret.
Response:
Thanks very much for reviewer’s comment. After this article was revised according to the reviewer's suggestion, we then submitted it to MDPI English editing service for English editing, in order to polish up language and style of this article.
- Methods must be reviewed extensively because not enough information is provided to understand the results.
Response:
Thanks very much for reviewer’s comment. We have detailed explanations in ‘Materials and Methods’ section, such as experimental procedures, sample preparation, quality control, software, parameters, references. Supplementary descriptions are added for method. Extensive modification and correction have been made in the new manuscript (in ‘Materials and Methods’ section).
- Please review the methods and results sections. The results section contains information about methods, cut-off values, and software that is not listed in methods.
Response:
Thanks very much for reviewer’s comment. The method-related statements have been removed from results section, which have been added to the ‘Material and Methods’ section. (see new manuscript Lines 78-150)
- There are several programs, software, and database that are not properly cited.
Response:
Thanks very much for reviewer’s comment. We have corrected the software version, the abbreviation and full name of the database name, and added some references. (see new manuscript Lines 92-97, 102-103, 116-118)
- The conclusion of the main manuscript is not understandable because of the writing style.
Response:
Thanks very much for reviewer’s suggestion. Original ambiguous expressions are deleted, and the ‘Conclusion’ section has been revised a lot. (see new manuscript Lines 439-445)
- It is not clear how authors defined phylogenetic clades.
Response:
Thanks very much for reviewer’s comment. Phylogenetic clades are defined following references written by Fang et al. (Int J Syst Evol Microbiol 2018;68:1146–1152) and written by Gabriel et al. (Appl. Environ. Microbiol. 2014, 80, 5359-5365).
- It is not clear how samples were used for transcriptome sequencing. More information should be provided about this procedure. There is no information in the main manuscript about the number of replicates, the number of reads by replicate, how the data was curated, and how reads were filtered.
Response:
We have added more information of transcriptome sequencing in our new manuscript (Lines 89-97):
- Three RNA samples of low-salt and another three RNA samples of high-salt were extracted, and each sample was sequenced for transcripts, which means 3 replicate measure in each condition. These states have been added in the text, see new manuscript Line 89. b. For data quality control, the reads were mapped to the genome. The reference genome is available in GenBank whose accession number is ‘GCF_002749895.1_ASM274989’. The average mapping rate of the ‘sample-genome alignment’ is 97.42%. c. Partial data collation was entrusted to BGI (Shenzhen, China) as assistant curate. d. To filter raw data, we use the SOAPnuke software for removing the adapters and removing those low-quality reads in the sequencing data. Reference titled ‘The Sanger FASTQ file format for sequences with quality scores, and the Solexa/Illumina FASTQ variants.’ by Cock P., et al. has been cited.
- I encourage authors to perform Average Nucleotide Analyses (ANI), Average Aminoacid Analyses (AAI), and in silico DNA-DNA hybridization analyses to obtain metrics about genome sequence identity. These analyses and metrics are important when performing comparative genomics, and also for genotaxonomy circumscription.
Response:
Thanks very much for reviewer’s valuable suggestion. The in silico DNA-DNA hybridization (is DDH) analysis is powerful in the genotaxonomy circumscription. The is DDH result was added in the new manuscript Lines 147-150. And we got the consistent results from AAI & ANI analyses.
Minor comments
Here I provide a list of suggestions for improving the manuscript. However the manuscript must be reviewed by an English editor before publication.
- Lines 14, 33, 36, 39, 41, 42, 81, 114, 120, 174, 196: Please change the genus name “Vibrios” to Vibrio. Vibrios is incorrect!.
Response:
Thanks very much for pointing out the mistake. All ‘Vibrios’ in this article have been replaced by word ‘Vibrio’. (see new manuscript Line 14, 38, 41, 45, 47, 48, 56, 158, 169, 201, 242, 268, 326, 372, 374)
- Line 22: aberrantly expressed. Please change the word aberrantly.
Response:
Thanks very much for pointing out the inappropriateness. Word ‘aberrantly expressed’ has been changed to word ‘differentially expressed’ (see new manuscript Line 22)
- Lines 24-25: “Notably, all differentially expressed genes significantly enriched in ‘structural constituent of ribosome’ and ‘ribosome’, which had considerable influence on salt press tolerance.”. Please re-write this sentence for clarity.
Response:
Thanks very much for pointing out the inappropriateness. This sentence has been changed to ‘Notably, differentially expressed genes have a significant association with ribosome structural component and ribosome metabolism, which may play a role in salt tolerance.’ (see new manuscript Lines 24-26)
- Line 26-27: “Transcriptional changes in ribosomal genes mean that V. fujianensis species may be a predominant advantage in a changing environment”. Please re-write this sentence for clarity.
Response:
Thanks very much for pointing out the inappropriateness. This sentence has been changed to ‘Transcriptional changes in ribosome genes mean that V. fujianensis may have gain a predominant advantage to adapt to the changing environment.’ Correction has been made in the new manuscript Line 28-30.
- Line 28: “has experienced extremely genome exchange events”. Please correct this phrase.
Response:
Thanks very much for pointing out the inappropriateness. This sentence has been changed to ‘In conclusion, to survive in adversity, V. fujianensis has enhanced its environmental adaptability and developed various strategies to fill its niche.’ (see new manuscript Lines 30-33)
- Line 33: “The genus Vibrios are ubiquitous and abundant”. -Vibrio-. Please correct this phrase.
Response:
Thanks very much for pointing out the mistake. Word ‘Vibrios’ has been replaced by word ‘Vibrio’. (see new manuscript Line 38)
- Line 35-36: “novel species scientifically recognized”. The -scientifically recognized- is not clear.
Response:
Thanks very much for pointing out the inappropriateness. Word ‘scientifically recognized’ has been changed to ‘More and more novel species have been scientifically identified, with more than 130 Vibrio species reported to date.’. (see new manuscript Lines40-42)
- Line 38: parahaemolyticus. Please include the full scientific name the first time you name a different species.
Response:
Thanks very much for pointing out the inappropriateness. Word ‘V. parahaemolyticus’ has been changed to ‘Vibrio parahaemolyticus’. (see new manuscript Line 44)
- Lines 47-49: “Secondly, Vibrios develop the adaptation strategies to survive extremes of salinity and temperature, through mechanisms of osmotic balance in their cytoplasm, modification of the lipid composition, activity of ion pumps and increase of the secondary metabolite production, et al. [9].”. Please re-write this phrase.
Response:
Thanks very much for pointing out the inappropriateness. This sentence has been changed to ‘Secondly, Vibrio spp. develop the adaptive strategies to survive in extreme condition of salinity stress and temperature, for instance, the mechanism of osmoregulation and osmotic balance, modification of the lipid composition, activity of ion pumps and increasing of the secondary metabolite production [9].’ (see new manuscript Lines 56-59)
- Line 52: “V. fujianensis”. Please include the full scientific name the first time you name a different species.
Response:
Thanks very much for pointing out the inappropriateness. Word ‘V. fujianensis’ has been changed to ‘Vibrio fujianensis’ (see new manuscript Line 62).
- Line 59-60: “In the present study, V. fujianensis as a representative species, comparative genomics was performed based on draft genomes”. Please re-write this phrase.
Response:
Thanks very much for pointing out the inappropriateness. This sentence has been corrected as ‘In the present study, a comparative genomics analysis was performed based on the V. fujianensis draft genomes and other reference genomes of genus Vibrio.’ (see new manuscript Lines 70-72).
- Line 60: “salt press conditions”. Please correct to clarify.
Response:
Thanks very much for pointing out the mistake. This sentence has been changed to ‘two salt stress conditions’ (see new manuscript Line 72).
- Line 66: “Strain V. fujianensis FJ201301T”. Please change to V. fujianensis FJ201301T or V. fujianensis strain FJ201301T in the whole manuscript.
Response:
Thanks very much for pointing out the inappropriateness. ‘Strain V. fujianensis FJ201301T’ in the whole manuscript have been changed to ‘V. fujianensis FJ201301T’. (see new manuscript Line 79, 120, 157, 163, 329, 402)
- Lines 71-73: “We had sequenced the whole genome sequence of V. fujianensis FJ201301T (Accession Number: GCA_002749895.1), and other 16S rRNA reference sequences (Table S1) closely related to taxa of the genus Vibrio were available in the GenBank database”. This phrase needs to be re-writted.
Response:
Thanks very much for pointing out the inappropriateness. It is right to delete word ‘We had’. This sentence has been corrected as “The whole genome sequence of V. fujianensis FJ201301T (accession number: GCA_002749895.1) was sequenced previously……are available in the GenBank”. Correction has been made in the new manuscript Lines 84-85, 100-101.
- Lines 73-74: “1% and 8% (NaCl, w/v) concentrations were chosen to represent low-salt stress and high-salt stress”. Please re-write for clarity.
Response:
Thanks very much for pointing out the inappropriateness. This sentence has been changed to ‘In order to represent low-salt stress and high-salt stress, 1% and 8% (NaCl, w/v) concentrations were chosen respectively.’ (see new manuscript Lines 85-88).
- Lines 76-77: “with an average yield of 10.52 Mb raw data per sample, then filtered reads were compared to the reference genome by HISAT 2.1.0 software”. Please include more information about the samples, and about how you filtered and processed reads. Please include all references used.
Response:
Thanks very much for reviewer’s comment. The raw reads were mapped to the genome for data quality control. To filter raw data, we use the SOAPnuke software for removing the adapters and removing those low-quality reads in the sequencing data. Reference titled ‘The Sanger FASTQ file format for sequences with quality scores, and the Solexa/Illumina FASTQ variants.’ by Cock P., et al. has been cited. (see new manuscript Lines 139-143).
- Line 79: “After a gene-prediction in the coding sequences”. Please correct
Response:
Thanks very much for pointing out the inappropriateness. Correction has been shown in the new manuscript Lines 107.
- Line 81: “Strain V. fujianensis FJ201301T”. Please change.
Response:
Thanks very much for pointing out the inappropriateness. Word ‘Strain V. fujianensis FJ201301T’ has been changed to ‘V. fujianensis FJ201301T’. Correction has been shown in the new manuscript Lines 107.
- Line 82: “included into evolution analysis of single-copy homologous genes”. Please re-write for clarity.
Response:
Thanks very much for pointing out the inappropriateness. This sentence has been modified as ‘Based on single-copy gene sequences and is DDH, V. fujianensis FJ201301T and other Vibrio reference genomes (Table S2) were included in the phylogenetic analyses.’ (see new manuscript Lines 119-121).
- Line 82-83: “Genes with a single-copy characteristic in each strain was identified as core genes”. This definition is not clear, please re-write.
Response:
Thanks very much for pointing out the inappropriateness. Here, we cite a reference written by Fabini et al., and statement has been changed as ‘Following the definition of single-copy core gene by Fabini et al. [15], genes with a single-copy characteristic in each strain was were identified as the core genes.’ (see new manuscript Lines 105-107).
- Line 83-84: “Maximum likelihood phylogenetic tree was reconstructed by MEGA (version 7.0) software”. Please include the reference for MEGA, and further explain how you perform this analyses. For instance, what was the selected substation model? How the tree is supported?, no branch-support values are shown.
Response:
Here we cite reference for MEGA which is written by Kumar S et al., and the genetic distance and sequence similarity of the 16S rRNA gene were calculated by MEGA software using the Kimura’s 2-parameter model. When constructing the phylogenetic tree, Kimura ’s 2-parameter model was selected as the alternative model; the pairwise-deletion option was used for the vacancy processing principle; 1000 bootstrap replicates is necessary for Bootstrap Test in phylogenetic tree. The 16S gene sequence phylogenetic tree in the results has robust bootstrap values, most are greater than 70%. But considering that the graph is not enough to accommodate all of the bootstrap values, it is not shown. Correction has been made in the new manuscript Lines 102-104.
- Line 87: “V. cincinnatiensis”. Please include the complete scientific name.
Response:
Thanks very much for pointing out the inappropriateness. Word ‘V. cincinnatiensis’ has been corrected to ‘Vibrio cincinnatiensis’ (see new manuscript Line 120).
- Line 91: “Web-BLAST (https://blast.ncbi.nlm.nih.gov/Blast) was applied to inferred”. Please include a reference for Web-BLAST.
Response:
Here, we cite a reference for BLAST-Web server which is written by Mount et al. (see new manuscript Line 128).
- Line 98: “FPKM (Fragments Per Kilobase of transcript per Million fragments mapped)”. I would suggest changing the order: first include the full name and then the abbreviation on parenthesis.
Response:
Thanks very much for good suggestion. This sentence has been changed to ‘The Fragments Per Kilobase of transcript per Million fragments mapped (FPKM)’ (see new manuscript Line 139)
- Line 99: “to calculate the transcript expression levels”. Please provide further information in the method section about this procedure.
Response:
Thanks very much for reviewer’s comment. In our study, Bowtie2 software was used to align clean reads to the reference genome sequence, and RSEM software was used to calculate the gene expression level of each sample. RSEM is a software package for RNA-seq reads to calculate gene and transcript subtype expression. Related references have been cited. (see new manuscript Line 138)
- Line 99: “FDR (false discovery rate)”. I would suggest changing the order: first include the full name and then the abbreviation on parenthesis.
Response:
Thanks very much for good suggestion. This word has been changed to ‘false discovery rate (FDR)’ (see new manuscript Line 141)
- Line 95: “VFDB”. I would suggest to include the full name for this abbreviation for clarity.
Response:
Thanks very much for suggestion. Correction has been made in the new manuscript Line 135.
- Lines 95-96: “To obtain high confidence of virulence related genes, VFDB blast cutoff value was set as the identity 90% with coverage cutoff 80%, E-value of 1e-5.”. Please re-write this phrase for clarity.
Response:
Thanks very much for pointing out the inappropriateness. This sentence have been corrected as ‘To obtain a high confidence of virulence related genes, the cutoff value was set as identity 90% and coverage 80%’. Correction has been made in the new manuscript Lines 135-136.
- Line 97: “(DEGs)”. I would suggest to avoid including an abbreviation in a subtitle, and better include it in the text.
Response:
Thanks very much for comment. We have deleted this abbreviation in the subtitle. (see new manuscript Line 148)
- Line 101: DEGs were defined. Here you can include the complete name and the the abbreviation.
Response:
We have added this abbreviation and complete name. This sentence has been changed to ‘In a given RNA-sequencing library, the DEGs were defined with a cutoff P-value ≤0.001 and a ≥2-fold-change compared between two samples.’ (see new manuscript Lines 143-145, a reference has been cited.)
- Lines 104-105: “The DEGs were annotated against the Swiss-Prot, GO, and KEGG databases by BLAST+ with a cut-off E-value of 10-5”. Please re-write this phrase for clarity.
Response:
This sentence have been changed to ‘The DEGs were annotated against the Swiss-Prot, Gene Ontology (GO), and Kyoto Encyclopedia of Genes and Genomes (KEGG) databases by BLAST+ with an E-value cutoff of 1e-5.’ Correction has been made in the new manuscript Lines 147-149. References have been cited.
- Line 109: “The 16S rRNA gene sequences was determined”. Please correct “sequences was”.
Response:
Thanks very much for pointing out the mistake. Correction has been shown in the new manuscript Line 191.
- Lines 109-110: “those 119 sequences of the genus Vibrio bacteria and strain Aeromonas hydrophila ATCC 7966T”. Please specify which 119 sequences.
Response:
Thanks very much for reviewer’s comment. Those 119 sequences have been already mentioned in the ‘Materials and Methods’ and have been listed in table S1. (see new manuscript Lines 101, 185)
- Lines 111-112: “Vibrio genus can be approximately divided into seven different clades as the phylogenetic tree demonstrated (Figure 1A)”. You need to re-write this phrase for clarity.
Response:
Thanks very much for reviewer’s comment. This sentence has been corrected as ‘Vibrio genus should be divided into seven different clades’ (see new manuscript Lines 155-156).
- Line 114: V. metschnikovii, V. bivalvicida, V. salilacus. Please include full scientific names.
Response:
Word ‘V. metschnikovii, V. bivalvicida, V. salilacus.’ Has been corrected as ‘Vibrio metschnikovii, Vibrio bivalvicida, Vibrio salilacus.’ (see new manuscript Lines 159-160).
- Lines 116-117: “Single-copy homology based phylogenetic tree (Figure 1B) showed that 10 strains were apparently divided into three main groups, group I to group III, respectively”. Please re-write this phrase.
Response:
Thanks very much for reviewer’s comment. Correction has been made in the new manuscript Lines 161-163.
- Line 118: “strain V. fujianensis FJ201301T”. Please write the scientific name properly.
Response:
Thanks very much for reviewer’s comment. Here, ‘strain V. fujianensis FJ201301T’ has been corrected as ‘V. fujianensis FJ201301T’.
- Lines 118-119: “had the closest evolutionary relationship with V. cincinnatiensis NCTC 12012T, implying these two species may occupy the similar functional ecological niche”. Please further explain this sentence. It is not clear how you can infer that if these two species are closely related they may have the same functional ecological niche. This must be reviewed.
Response:
Thanks for your useful and constructive comment. Here we do exaggerate the "imply" content, which has caused some confusion and misunderstanding. We have done a detailed review. This sentence has been corrected as ‘At the genome level, V. fujianensis FJ201301T was more closely related to V. cincinnatiensis NCTC 12012T, implying that they probably shared common ancestors in the past, according to the phylogenetic relationship’. Correction has been made in the new manuscript Lines 163-166.
- Lines 119-121: “Compared with the 16S rRNA gene phylogenetic tree, except for V. salilacus DSG-S6T, the remaining Vibrios of the Cincinnatiensis clade still formed a distinct cluster relatively similar to V. cholerae strains cluster”. Please re-write this phrase.
Response:
This sentence has been corrected as ‘Compared with the 16S rRNA gene phylogenetic tree, genetic relationship is slightly different. Here V. salilacus DSG-S6T separated from Cincinnatiensis clade, while the remaining Cincinnatiensis clade Vibrio spp.; still gathered closely, which were divided into the group III together with V. cholerae strains’ (see new manuscript Lines 167-170).
- Line 123: V. fujianensis. Please include the full scientific name.
Response:
We have added this full scientific name ‘Vibrio fujianensis’ in the text. (see new manuscript Line 180).
- Lines 123-124: “Phylogenetic tree among the genus Vibrio based on 16S rRNA gene ---sequences---. “(B) Phylogenetic tree of the single-copy homology”. This figure title is not clear.
Response:
Thanks very much for pointing out the inappropriateness. This sentence has been corrected as ‘Phylogenetic tree among the genus Vibrio based on 16S rRNA gene sequences’. (see new manuscript Line 181).
- Line 127: “The accession number of 16S rRNA gene and genomes were shown”. Please correct number(s) gene (sequences).
Response:
Thanks very much for pointing out the inappropriateness. This sentence has been corrected as ‘The accession numbers of 16S rRNA gene sequences and genomes were are shown in Tables S1 and S2, respectively”. (see new manuscript Lines 184-185).
- Lines 132-134: “An additional gene set ranged from 178 to 1881 were shared by any two species, while any three species had co-ownership of range from 82 to 882 genes together”. Please re-write for clarity. What do you mean by “co-ownership of range”. Please correct.
Response:
Thanks very much for reviewer’s comment. Venn diagram showed that the overlapping area represented the number of genes shared by any two or three species. This sentence has been corrected as ‘An additional gene set, ranging from 178 to 1881, was shared by any two species, while the number of genes were shared by any three species varied from 82 to 882.’ Correction has been made in the new manuscript Lines 195-197.
- Lines 136-137: “In terms of quantity, each species universally contained more than 3,000 core genes, and V. cholerae MO45 was slightly higher than that of the other three Vibrios”. Please re-write this sentence.
Response:
Thanks very much for reviewer’s comment. This sentence has been corrected as ‘In terms of quantity, each Vibrio species contained more than 3,000 core genes, and V. cholerae MO45 had slightly more genes than other three bacteria’ (see new manuscript Lines 200-201).
- Line 130: V. metschnikovii. Please include the full scientific name.
Response:
Thanks very much for reviewer’s comment. The word ‘Vibrio metschnikovii’ has appeared in the line 159. Here is not for the first time.
- Line 135: “Comparison of the number of core genes were calculated among genomes”. Please correct this sentence.
Response:
Thanks very much for reviewer’s comment. This sentence has been corrected as ‘The number of core genes were calculated from the genomes mentioned above’. Correction has been made in the new manuscript Line 199.
- Lines 138-140: “On the other hand, the decreasing trend (Figure 2C) had become more apparent as the number of Vibrio species included in the analysis increases”. Please re-write this sentence.
Response:
Thanks very much for reviewer’s comment. This sentence has been corrected as ‘On the other hand, the decreasing trend (Figure 2C) became more apparent when the number of strains continued to increase’. Correction has been made in the new manuscript Lines 203-205.
- Line 142: “Comparative genomic analysis of V. fujianensis”. The figure includes information for several Vibrio strains not only V. fujianensis. Please correct.
Response:
Thanks very much for reviewer’s suggestion. This sentence has been corrected as ‘Comparative genomic analysis of V. fujianensis and three other Vibrio species’ (see new manuscript Line 207).
- Lines 143-144: “The shared and unique genome among compared genomes”. Please re-write for clarity.
Response:
Thanks very much for reviewer’s comment. This sentence has been deleted, and added into ‘Materials and Methods’ section. Correction has been made in the new manuscript Line 114.
- Lines 148-149: “(C) Core genes dilution curves in the genomes”. Please re-write this sentence. It is not understandable.
Response:
This sentence has been corrected as ‘(C) Core gene quantitative trend’ (see new manuscript Lines 214).
- Line 153: “strain FJ201301T”. Please include the full scientific name.
Response:
Word ‘strain FJ201301T’ has been corrected as ‘V. fujianensis FJ201301T’. (see new manuscript Line 219).
- Line 160: Mannose. Change to mannose.
Response:
Thanks very much for pointing out the inappropriateness. Word ‘Mannose’ has been changed to ‘mannose’. (see new manuscript Line 227)
- Line 161: Asparagine. Change to asparagines.
Response:
Thanks very much for pointing out the inappropriateness. Word ‘Asparagine’ has been changed to ‘asparagines’. (see new manuscript Line 228)
- Lines 180-181: “We had predicted the virulence-associated factors of the V. fujianensis FJ201301T genome and common pathogens against virulence factors database (web-VFDB).”. Please re.write this sentence and provide a reference for the database.
Response:
This sentence has been corrected as ‘The virulence-associated factors of V. fujianensis FJ201301T and other common pathogens were predicted against VFDB’. (see new manuscript Lines 272-273) As for the reference for VFDB, it has been cited in ‘Materials and Methods’.
- 182: “kinds of virulence genes”. How do you define a “kind” of virulence gene?.
Response:
Here the word ‘kind’ refers to type of virulence-related gene, and that is, different virulence-related factors and virulence-related genes are provided derictly by VFDB based on the prediction results. Then we counted the number of different genes that are the types of virulence-related genes.
- Table 1: V. fujianensis. Please include the full scientific name.
Response:
Thanks very much for suggestion. Table 1 ‘V. fujianensis’ has been corrected as ‘V. fujianensis FJ201301T’. (see new manuscript Line 287).
- Line 196: VFDB. Please include the full name of the abbreviation for clarity.
Response:
Thanks very much for suggestion. The full name of VFDB has been added in the text. (see new manuscript Line 290).
- Line 198: “R 3.5.2 software”. Please correct and properly cite this software.
Response:
Thanks very much for pointing out the inappropriateness. Word ‘R 3.5.2 software’ has been corrected as ‘R-3.5.2’. And it has been cited in ‘Materials and Methods’. (see new manuscript Line 116)
- Line 201: “genes involved in two salt tolerance”. Please correct.
Response:
This sentence has been corrected as ‘genes involved in salt tolerance’. (see new manuscript Line 273)
- Lines 202-203: “The length distribution of the transcriptome”. Please re-write this phrase. It is not clear what do you mean by “length distribution of the transcriptome”.
Response:
Thanks very much for suggestion. This sentence ‘The length distribution of the transcriptome’ has been corrected as ‘The length distribution of the transcripts’. (see new manuscript Line 275)
- Line 204: The box plot. Please provide further information. A box plot of what?
Response:
Thanks very much for comment. This sentence has been changed to ‘A box plot of the total gene expression’ (see new manuscript Line 277)
- Line 206: “aberrantly expressed”. Please change aberrantly.
Response:
Word ‘aberrantly expressed” has been changed to ‘differentially expressed’. (see new manuscript Line 280)
- Lines 211-213: “Transcriptome gene expression analysis. (A) box plot of total gene expression. (B) volcano plot of gene expression. The blue and red color indicate up-regulated and down-regulated genes, respectively”. Please re-write this phrase.
Response:
Thanks very much for comment. This sentence has been changed to ‘Gene expression analysis. (A) Box plot of the total gene expression. (B) Volcano plot of gene expression. The blue and red color represent up-regulated and down-regulated genes, respectively’. Correction has been made in the new manuscript Lines 284-286.
- Lines 237-238: “The ribosome, one of the most important organelles, is described as a plant for protein biosynthesis in cells and has had considerable influence on salt press tolerance”. Please re-write the phrase. The “one of the most important organelles” can be omitted. “described as a plant for protein biosynthesis in cells” what do you mean by plant, and please provide a reference to support this phrase.
Response:
Thanks very much for reviewer’s comment. We removed the phrase ‘one of the most important organelles.’ Line 237 ‘plant’ means protein processing factory, plant does not refer to trees and flowers. This may be a misunderstanding caused by polysemy in English. In consistence with Reviewer #1, comments (14), this word has been deleted. Correction has been made in the new manuscript (line 311).
- Lines 241-243: “Transcriptional changes in ribosomal genes mean that V. fujianensis species respond more efficiently and quickly to high-salt press, which may be a predominant advantage in a changing environment”. Please re-write for clarity and please provide further information to support this claim.
Response:
Thanks very much for reviewer’s comment. This sentence has been corrected as ‘Transcriptional changes in the ribosomal genes indicated that V. fujianensis species responded efficiently and quickly to high-salt stress, which might be a predominant advantage in the changing environment’. Correction has been made in the manuscript (line 316-318).
- Lines 247-248: “Candidate unigenes related to salt press. Red represents up-regulated, while green represents down-regulated”. Please re-write and provide further information to interpret the figure
Response:
Thanks very much for reviewer’s comment. Correction has been made in the manuscript (line 322-323).
- Lines 267-268: “We found that great different in the representative genomes”. Please correct this sentence.
Response:
Thanks very much for reviewer’s comment. This sentence has been modified as ‘Great difference in the representative genomes of …’. Correction has been made in the new manuscript (line 345).
- Lines 269-271: “which had significant influence on proportion of core genes and accessory gene, consequencely produced diverse environmently-adaptive phenotypes depending on the genotype [26]”. Please re-write for clarity.
Response:
Thanks very much for reviewer’s comment. This sentence has been re-written as ‘Genotype determines phenotype, and consequently, diverse phenotypes related to the environmental adaptability are present or absent during the evolution’. Correction has been made in the manuscript (line 348-350).
- Lines 274-275: “might confer body capability of being a strong and secure super organism through environmental-troubled times”. This phrase is out of context, please re-write or further explain it.
Response:
Thanks very much for reviewer’s comment. This sentence has been re-written as ‘These strain-specific genes with an unequal number and diverse function might confer the microbe some potential to go through environmental-troubled times.’ Correction has been made in the manuscript (line 353-356).
- Lines 331-332: “To our knowledge, microorganisms form a variety of physiological mechanisms and invoke biological strategies that adapt to high salt environment”. Please re-write and provide references for this sentence.
Response:
Thanks very much for reviewer’s comment. This sentence has been re-written as ‘To our knowledge, most microorganisms form a variety of biological mechanisms and physiological responses, and those mechanisms can help them take advantage of high salt environments’. The references have been cited. Correction has been made in the manuscript (line 424-427).
- Lines 333-334: “On the one hand, they rely on a salt rejection strategy for compatible solute accumulation, which usually contains polarizable chemical substances. Please re-write for clarity and please explain what do you mean by polarizable chemical substances.
Response:
Thanks very much for reviewer’s comment. The types of organic molecules used for osmotic balance include polyols and derivatives, sugars and derivatives, amino acids and derivatives, betaines, and ectoines and occasionally peptides suitably altered to remove charges. All these were listed in the reference written by Mary F Roberts. And we cited this reference in the manuscript (line 428-433).
- Lines 286-288: “By assembling surface antigen structure [33] of bacteria, secretion of virulence effect protein [34], and other ways to arm itself, so as to escape the immune defense system in host, V. fujianensis has strong adaptability to a new environment”. Please re-write.
Response:
Thanks very much for reviewer’s comment. This sentence has been modified as ‘bacteria become so powerful so as to escape the immune defense system in host. Likewise, V. fujianensis has a strong adaptability to a new environment.’ Correction has been made in the manuscript (line 369-371).
- Lines 289-292: “It carries a considerable number of virulence-associated factors in pathogenic Vibrios virulence-gene pool (Table 1, Table S3), indicating that there are extensive gene communication and gene transfer between V. fujianensis and other Vibrios in the water environment, and one bacterium donates genetic material to another”. Please re-write for clarity.
Response:
Thanks very much for reviewer’s comment. This sentence has been re-written as ‘It carries a number of virulence-associated factors in the virulence-gene pool among pathogenic Vibrio (Tables 2 and S3), indicating that there may be an extensive gene swap or gene exchange between V. fujianensis and other pathogenic Vibrio. There is reason to believe that any two major groups of micro-organisms can share their genetic codes’. Correction has been made in the manuscript (line 371-376).
- Line 301: we find. Please change to we found.
Response:
Thanks very much for reviewer’s comment. Correction has been made in the manuscript (line 345).
- Line 302: “This finding well explains the phenomenon”. Please correct the sentence.
Response:
Thanks very much for reviewer’s comment. This sentence has been re-written as ‘This finding was a reasonable explanation for the cross agglutination between V. fujianensis and V. cholerae O139 serum’. Correction has been made in the manuscript (line 388).
- Lines 326-327: “The decrease in ribosome biogenesis and translation in C. albicans [48] by proteomic analyses was also statistically significant”. This sentence is out of context please provide more information or connect it better to the context of the manuscript.
Response:
Thanks very much for reviewer’s comment. This sentence has been re-written as ‘Proteomic analyses in C. albicans revealed a link between ribosomal gene expression and environmental adaptation [48]’. Correction has been made in the manuscript (line 415).
- Lines 327-328: “The discovery of the high-resolution structure of the ribosome, especially the key active sites, is decisive for salt tolerance”. Please re-write for clarity and provide further information, support and references for this claim.
Response:
Thanks very much for reviewer’s comment. This sentence has been replaced by ‘Salt tolerance is a common feature of Escherichia coli cells, in which the maturation or function of the ribosome are impaired. Changes in the ribosome confer salt resistance on microbes. Microbes begin to synthesize or uptake osmo-protectants in high salt condition, while inhibiting general σ70 transcription’. Correction has been made in the manuscript (line 417-420). References have been cited.
- Lines 328-330: “Na+/H+ antiporter of Vibrio has effect on survival in salt stress environment, and NhaB operates is proved to be a possible mechanism of regulating Na+/H+ antiporter activity [49]”. Please re-write for clarity.
Response:
Thanks very much for reviewer’s comment. This sentence has been re-written as ‘The Na+/H+ antiporter makes it effective for the survival of Vibrio species in a saline environment. NhaB is proven to be a possible mechanism for regulating Na+/H+ antiporter activity’. Correction has been made in the manuscript (line 421-423).
Reviewer 4 Report
The study by Huang et al. describes a new Vibrio species, Vibrio fujianensis. The authors study the environmental response of V. fujianensis at low (1% w/v) and high (8% w/v) salt press conditions by RNA-sequencing and identifying differentially expressed genes.
The study is well described, analyses are sound, and figures are clear. I have the following comments:
- Add how RNA-sequencing libraries were prepared.
- What software was used to identify Differentially Expressed Genes?
- Check singular/plural forms of verbs and typos throughout the manuscript. For instance:
- title: "analysis" is singular but the verb is at the plural form;
- line 109: "sequences" is plural and the verb is at the singular form;
- line 341: is "might" missing?
Author Response
Reviewer #4 (Comments for the Author):
The study by Huang et al. describes a new Vibrio species, Vibrio fujianensis. The authors study the environmental response of V. fujianensis at low (1% w/v) and high (8% w/v) salt press conditions by RNA-sequencing and identifying differentially expressed genes.
The study is well described, analyses are sound, and figures are clear. I have the following comments:
- Add how RNA-sequencing libraries were prepared.
Response:
Thanks very much for reviewer’s comment. We added “The whole RNA-sequencing process, including RNA library construction, sequencing, and data pipelining, was done in accordance with the manufacturer’s protocols by a commercial sequencing service (BGI, Shenzhen, China).” in our new manuscript. Lines 94-97. (Track Changes are shown under Final Showing Markup)
- What software was used to identify Differentially Expressed Genes?
Response:
Thanks very much for reviewer’s comment. Differentially Expressed Genes (DEGs) were identified using PossionDis method which is based on the Poisson distribution principle. This method has been published on Genome Research by Audic S. et al. At the same time, the definition of DEGs has been mentioned in ‘Materials and Methods’. (see new manuscript Lines 139-145)
- Check singular/plural forms of verbs and typos throughout the manuscript. For instance:
title: "analysis" is singular but the verb is at the plural form;
line 109: "sequences" is plural and the verb is at the singular form;
line 341: is "might" missing?
Response:
Thanks very much for pointing out the inappropriateness.
The word ‘analysis’ changed to ‘analyses’.
Line 109 has been corrected as ‘The 16S rRNA gene sequences of V. fujianensis FJ201301T was determined’.
Line 341 has been added word ‘might’.
Correction has been made in the new manuscript (title, Lines 153, 440).
Round 2
Reviewer 3 Report
Authors considered all comments and suggestions.
The manuscript has improved and has more scientific soundness.
However, there are still sentences that should be reviewed for English language and style. Most of these sentences are the ones you included in this new version of the manuscript. The rest of the original manuscript (first version) is now more readable.
Also, I suggest authors to review and re-write your conclusion, as it is still confusing.
These are my last comments and suggestions to improve the manuscript:
Lines 202-204: “The phylogenetic analysis for the genotaxonomy circumscription was performed using an in silico DNA–DNA hybridization (is DDH) web server called the genome-to-genome distance calculator…”. Please write in silico DDH not “is DDH” it is not useful for reading with the abbreviation of in silico, through all of the manuscript. You can skip the “in silico” an only refer to “DDH values” Also, DDH values are useful for genotaxonomy circumscription because they provide quantitative metrics for genome comparisons, not for phylogenetic analyses. In this regard, genome-based metrics allow a more accurate taxonomic circumscription of bacterial species. So please re-write this sentence. “DNA-DNA hybridization (DDH) values were computed using the Genome-to-Genome Distance Calculator v.2.1 (8).”
Lines 204-206: “Based on single-copy gene sequences and is DDH, V. fujianensis FJ201301T and other Vibrio reference genomes (Table S2) were included in the phylogenetic analysis”. Please re-write for clarity.
Line 207: “2.4. O-Polysaccharide (O-PS) Analysis”. Please include a more descriptive subtitle: “2.4. O-Polysaccharide (O-PS) --[gene/gene cluster(s)]-- Analysis.
Lines 208-209: The O-PS --[gene sequence(s)/gene cluster sequence]-- of V. fujianensis FJ201301T was aligned against the reference O-PS --[gene sequence(s)/ gene cluster sequence]-- in the genome of V. cholerae O139 serovar MO45. Please re-write.
Line 213: O-PS [gene cluster/ gene region] region. Please specify.
Line 214: “homologous regions in the O-PS”. Please specify. “homologous regions in the O-PS --[gene cluster(s)?]--”.
Line 215: “The visual comparison analysis of the homologous regions in the O-PS was performed…”. Please re-write for clarity. “A comparative analysis of homologous regions of Vibrio O-PS gene clusters/regions was performed… ”.
Lines: 234-236: “The 16S rRNA gene sequence of V. fujianensis FJ201301T was determined and compared to those 119 sequences of the genus Vibrio bacteria and Aeromonas hydrophila ATCC 7966 T which served as an outgroup”. Please re-write. You can split this sentence into two sentences. Maybe: “The 16S rRNA gene sequence of V. fujianensis FJ201301T was aligned and compared to a set of 119 corresponding sequences of other Vibrio species and strains. Aeromonas hydrophila ATCC 7966 T served as an outgroup species.”. Also, for phylogenetic analyses you must include -at least two- outgroup sequences in order to provide further support to the outgroup clade. In your phylogenetic analysis you only used one outgroup sequence. It is highly possible that the tree would not suffer major changes so, please take this as a recommendation for you to evaluate if you want to include a new tree with more outgroup sequences.
Line 337: “The is DDH”. Please change to: “The DDH value(s)”
Lines: 338-340: “The intergenomic relatedness values among these type strains were 18.70%–65.90% for is DDH, less than the proposed cutoff levels (70%) for species delineation..”. Please include the reference about the cutoff value of 70%. Please change “intergenomic relatedness values” to “DDH values”. Please change “cutoff levels” to “cutoff values”. Please change “less” to “lower”. “The DDH values among these type strains were 18.70%–65.90%. These DDH values are lower than the proposed cutoff value (70%) for species delineation..”.
Question for authors: From which formula did you report DDH values?. The web tool provides values for three different formulas. The second formula is the recommended one.
In the figure description: ”(B) Phylogenetic tree of the single-copy homology” include the word –genes-. “(B) Phylogenetic tree based on homologous gene sequences of Vibrio species analyzed in this study”.
Line 346: “All single-copy homologous genes --[for each species/strain]-- were concatenated to form a new sequence 97,365 bp in length”. Please specify.
Line 425: “3.3. Comparative Analysis of O-PS [genes/ gene regions/ gene clusters]”
Lines 426-427: “The O-PS [gene/gene region] clustering analysis (Figure 3A) showed that V. fujianensis FJ201301T had a large [sequence?gene array?] variation compared with V. metschnikovii JCM 21189T and V. cincinnatiensis NCTC 12012T. To what large variation do you refer? Do you have a sequence based comparison value maybe from a blast comparison?. Please specify.
Lines 429-431: “Not only is The O-PS --[gene sequence(s)/gene cluster sequence(s)]-- of V. fujianensis FJ201301T *is* similar to the *counterparts* of V. metschnikovii --[include strain name]-- and V. cincinnatiensis --[include strain name]—Do you have a sequence identity percentage value for the sequence(s)?. We found that the O-PS --[gene region/gene cluster] of V. fujianensis [include strain name] had three homologous fragments *shared with the same region/cluster of* V. cholerae MO45”.
Line 470: Comparative analysis of O-polysaccharide (O-PS) genes/gene regions/gene clusters.
In the figure legend: “Clustering of O-PS *genes/gene regions/gene clusters*”
Lines 470-471: O-PS *genes/gene regions/gene clusters* comparison(s) between V. fujianensis FJ201301T with other Vibrio species.
Lines 473-474: “Homologous fragment gene clusters (nucleotide site from 990 to 20158) of O-PS between V. fujianensis FJ201301T and V. cholerae O139 serovar MO45.”. Please re-write for clarity.
Line 523: “A clustering analysis? of pathogenic Vibrio *species* *is* shown in”. Or: “pathogenic Vibrio *species* are clustered and “are” shown in”.
Lines 600-602: “Figure 5. Gene expression analysis. (A) Box plot of the total gene expression. (B) Volcano plot of the gene expression.” Please specify: ““Figure 5. Gene expression analysis – of what?-. (A) Box plot of the total gene expression – of what?-. (B) Volcano plot of the gene expression – of what?-.”. Please provide a more descriptive figure legend for readers.
Line 725: “It has been shown that Vibrio is characterized”. “that the Vibrio genus is”, “that Vibrio species are”.
Lines 726-728: “In this study, the phylogenetic tree constructed by the based on 16S rRNA gene sequences divided of 119 Vibrio species *defined* seven phylogenetic clades [include a reference to the figure. Fig X]. *Vibrio* fujianensis FJ201301T *was related to the Cincinnatiensis clade.”
Lines 735-736: “These newly reported species (which one(s)?) have great reliability and increasing stability (to what are you meaning with great reliability and stability?This is confusing)”
“so as to make *define* the Cincinnatiensis clade separate *as independent?* from the Cholerae clade (they already have two different names, so this is confusing?), and to become an evolutionarily-independent clade in the phylogenetic tree.” Maybe: ““We propose that the Cincinnatiensis clade is an evolutionarily-independent Vibrio clade based on the phylogenetic tree.”
Please re-write these sentences for clarity.
Lines 740-741: “V. cincinnatiensis and V. metschnikovii were recognized as two members of V. fujianensis’s close relatives. Please re-write.
Lines 742-743: “Great difference in the representative genomes of the four Vibrio species (V. fujianensis, V. cincinnatiensis, V. metschnikovii, and V. cholerae) were found, which had a significant influence on some aspects, like the GC content, genome structure, proportion of core genes and accessory genes”. Include all strain names. *Great difference* in what aspect? Maybe great difference was found between representative genomes of…?. It is not clear how “the great difference” affects all of these: “significant influence on some aspects, like the GC content, genome structure, proportion of core genes and accessory genes”. Please re-write this sentence.
Lines 745-746: “Genotype determines phenotype, and consequently, diverse phenotypes related to the environmental adaptability are present or absent during the evolution [38].” I considered you can delete this sentence. Genotype and the environmental variables does affect phenotypes. This sentence is confusing and can be omitted.
Line 746: Each *Vibrio* strain
Lines 778-779: “These strain-specific genes with an unequal number and diverse function*s* might confer the microbe (which microbe?) some potential to go through environmental-troubled times”.
Lines 796-797: “There is reason to believe that any two major groups of micro-organisms can share their genetic codes”. This is not clear. Please delete or re-write.
Lines 799-800: “Mobile genetic elements, such as plasmid/phage (pro-phage)/integron, are found to be a crucial factor for horizontal gene transfer”. This is not clear. Please delete or re-write.
Line 806: “we found that the O-PS *genes/gene regions/gene clusters*”
Line 806: “V. fujianensis [include strain name] and V. cholerae MO45”
Lines 809-811: “With combined atypical GC content values, we speculated *propose* that between the V. fujianensis [include strain name or serotype] and V. cholera [include strain name] O139 serotype, a horizontal gene transfer of partial O-PS gene clusters *may have* occurred under certain circumstances (for example which circumstances)”. This sentence is written in a very speculative form. I suggest to re-write it.
Line 815: “More research shows that *the*O-antigen in Vibrio *species*”.
Line 816: “which is beneficial to develop alternative lifestyles during intestinal or organism–animal colonization.”. Please re-write for clarity. what are you meaning with organism–animal colonization?
Lines 817-819: “Therefore, we logically believed *propose/consider* that the exchange of partial O-antigen fragments between V. fujianensis *strain name* and V. cholerae O139 can greatly enhance the environmental adaptability of V. fujianensis *strain* to effectively breed and tenaciously survive in nature”. What do you mean with “to effectively breed”. Please-rewrite.
Lines 930-933: “Salt tolerance is a common feature of Escherichia coli cells, in which the maturation or function of the ribosome are is? Impaired (This sentence is not clear). Changes in the ribosome confer salt resistance on microbes (please include reference(s)). Microbes (which microbes?) begin to synthesize or uptake osmo-protectants in a high salt condition, while inhibiting general σ70 transcription”.
Lines 935-937: “To our knowledge, most microorganisms form a variety of biological mechanisms and physiological responses (to what?), and those mechanisms can help them take advantage of the high salt environments (all of the microorganisms?)”. This sentence is not clear.
Lines 948-950: “In this study, changes in the ribosome pathway may be related to salt tolerance via transcriptome sequencing, which implies that V. fujianensis has adapted to survive in a changing environment”. Please correct this sentence: “In this study, changes in the ribosome pathway may be related to salt tolerance via transcriptome sequencing”.
Lines 950-951: “Strain-specific genes in [the]? V. fujianensis /strain/species/clade? might help develop its innate adaptability, as well as its potential ability to evolve quickly and thrive in every ecological niche (all niches? To which?)”
Lines 951-952: “Several genetic motility genes might attribute to HGT in the evolutionary process”. Please re-write.
Supplementary information
Figure S1. Transcriptome length distribution of V. fujianensis. Please change to: Sequence length distribution of V. fujianensis transcripts analyzed in this study.
Figure S2. Cluster heatmap analysis of DEGs patterns. Please provide more information about the figure. What do colors represent? What represents the lines (phylogenetic clustering?). Please provide more information to readers about this figure.
Table S1. 16S rRNA gene based vibrios overview analyzed in this study. Please change to: Overview of 16S rRNA gene sequences of Vibrio species analyzed in this study.
Table S2: Genomic overview of Vibrio species analyzed in this study. Recommendation for this table: It would be very valuable to include the genome size and GC content values in two new columns.
Table S3. Virulence-associated factors profile of V. fujianensis and other pathogenic Vibrio species. Please write gene names in italics. Example: acfA = acfA.
